# Postnatal maternal depressive symptoms and behavioural outcomes in term-born and preterm-born toddlers: a longitudinal UK community cohort study

Ira Kleine ,[1] George Vamvakas,[2] Alexandra Lautarescu,[1,3] Shona Falconer,[1] Andrew Chew,[1] Serena Counsell,[1] Andrew Pickles,[2] David Edwards,[1] Chiara Nosarti[1,4]

For numbered affiliations see end of article.

**Correspondence to**
Dr Chiara Nosarti;
chiara.nosarti@kcl.ac.uk

## ABSTRACT

**Objectives** To examine the association between maternal depressive symptoms in the immediate postnatal period and offspring's behavioural outcomes in a large cohort of term-born and preterm-born toddlers.

**Design and participants** Data were drawn from the Developing Human Connectome Project. Maternal postnatal depressive symptoms were assessed at term-equivalent age, and children's outcomes were evaluated at a median corrected age of 18.4 months (range 17.3–24.3).

**Exposure and outcomes** Preterm birth was defined as <37 weeks completed gestation. Maternal depressive symptoms were assessed with the Edinburgh Postnatal Depression Scale (EPDS). Toddlers' outcome measures were parent-rated Child Behaviour Checklist 1½–5 Total (CBCL) and Quantitative Checklist for Autism in Toddlers (Q-CHAT) scores. Toddlers' cognition was assessed with the Bayley Scales of Infant and Toddler Development—Third Edition (Bayley-III).

**Results** Higher maternal EPDS scores were associated with toddlers' higher CBCL (B=0.93, 95% CI 0.43 to 1.44, p<0.001, $f^2$=0.05) and Q-CHAT scores (B=0.27, 95% CI 0.03 to 0.52, p=0.031, $f^2$=0.01). Maternal EPDS, toddlers' CBCL and Q-CHAT scores did not differ between preterm (n=97; 19.1% of the total sample) and term participants. Maternal EPDS score did not disproportionately affect preterm children with respect to CBCL or Q-CHAT scores.

**Conclusions** Our findings indicate that children whose mothers reported increased depressive symptoms in the early postnatal period, including subclinical symptoms, exhibit more parent-reported behavioural problems in toddlerhood. These associations were independent of gestational age. Further research is needed to confirm the clinical significance of these findings.

## INTRODUCTION

Postnatal depression affects approximately 12% of mothers worldwide.[1] In contrast to 'baby blues', which is a state of emotional lability that affects between 13.7% and 76.0% of women in the first few days after birth and typically resolves spontaneously within 2 weeks,[2] postnatal depression is more severe and starts in the first few months postpartum.[1] Stressful life events have been linked to a heightened risk of developing postnatal depression[3]; for example, mothers of preterm infants have a significantly higher risk of postpartum depression compared with mothers of term infants,[4] likely due to heightened stress associated with perinatal complications.[5]

Women with postnatal depression tend to be less responsive to their baby's needs and to display less affection.[6] Therefore, in the short-term, postpartum depression may affect mother–infant interactions[7] and in the long term, it may lead to alterations in brain development,[8] emotional difficulties,[9] less secure attachment, cognitive and behavioural problems in childhood and a possible increased risk of autism spectrum disorder (ASD).[10 11] Large cohort studies, such as the Avon Longitudinal Study of Parents and Children, have shown that these associations are even evident

---

### STRENGTHS AND LIMITATIONS OF THIS STUDY

⇒ Prospective study with a large sample, using multiple imputation to reduce non-response bias.
⇒ Maternal depressive symptoms assessed as a continuous variable, providing more nuanced information about the significance of subclinical symptoms.
⇒ Maternal depressive symptoms assessed earlier than in previous studies, enabling recognition of early screening opportunities for families.
⇒ Potential common method variance bias through parent-completed child behavioural assessments.
⇒ Unknown paternal and parental factors, such as comorbid psychiatric conditions, that may confound our findings.

when maternal depression is measured on a continuum of symptoms rather than a dichotomous diagnosis,[12–14] supporting the notion that elevated subdiagnostic psychiatric symptoms can also negatively impact on children's development.[15]

Studies investigating the underlying causes that may link maternal postnatal depression to child outcomes have implicated several biological and environmental variables. For instance, genetic and epigenetic factors have been shown to both mediate and mitigate the intergenerational transmission of psychiatric disorders,[16] while lower quality parenting, interparental conflict and socioeconomic deprivation have been shown to exacerbate children's developmental risk of emotional and behavioural problems.[11] In addition, being born preterm (ie, <37 weeks' gestation, as per the WHO definition)[17] has been associated with alterations in early brain development[18] as well as neurological, behavioural and cognitive problems in childhood and beyond.[19 20] Therefore, it is complex to disentangle the possible effects of postnatal maternal mental health and those of perinatal clinical factors on specific outcomes in preterm children, as these may involve both maternal psychosocial and biological variables, as well as child preterm-related neurodevelopmental morbidity.

Furthermore, a question that remains unanswered is whether preterm birth (PTB) accentuates the association between maternal postnatal depression and child outcome. Two theoretical frameworks exist that hypothesise certain infants may be influenced differently by external stimuli: the diathesis stress model proposes that certain vulnerability factors make affected infants more prone to suboptimal environmental influences with subsequent poorer outcomes,[21 22] whereas the differential susceptibility model frames such factors as plasticity mediating, thus leading to poorer outcomes in negative environments, as well as better outcomes in supportive environments.[22 23] Previous studies investigating differential susceptibility have shown mixed findings studying a range of environmental and clinical exposures,[24 25] with child outcomes including attachment, internalising and externalising behaviour and academic competence.[25] Both low birth weight in term infants (small for gestational age, SGA)[26] and PTB[23 24 27] have been explored as distinct potential susceptibility factors. This distinction is based on the different pathophysiological processes underlying the respective conditions of SGA and PTB, both, or a combination, of which can cause low birth weight.[28] For example, SGA is a marker of intrauterine growth restriction related to placental dysfunction,[29] whereas PTB can be caused by a multitude of factors, including infection and inflammation.[30]

Given that mothers of preterm children experience elevated levels of distress,[31] are at high risk of developing postnatal depression,[32] and that preterm children themselves are vulnerable to psychiatric sequelae,[33] in addition to investigating the association between very early maternal postnatal depressive symptoms and child behavioural and emotional outcomes, we further aimed to investigate the interaction between PTB and maternal depressive symptoms on child outcomes. Previous work focusing on the differential susceptibility of preterm born children to various environmental stimuli, as described above, had not yet studied maternal depressive symptoms as a proposed exposure. We specifically aimed to investigate the continuum of maternal depressive symptoms rather than solely focussing on clinically significant maternal depression, so as to provide more nuanced information about the importance of subclinical depressive symptoms on child outcomes. We hypothesised that early postnatal maternal depressive symptoms would be more elevated in mothers of preterm compared with term infants and that these would impact preterm children's behavioural and emotional outcomes to a greater degree than their term counterparts.

## METHODS
### Sample
Participants were enrolled in the Developing Human Connectome Project (DHCP, http://www.developing-connectome.org/), a neuroimaging-focused project, with eligibility criteria including pregnant women (aged ≥16 years) with a gestational age of 20–42 weeks, and newborn infants aged 24–44 weeks; infants enrolled in the DHCP had MRI at term-equivalent age. Exclusion criteria for the DHCP included: contraindications to MRI, babies being too unwell to tolerate a scan and language difficulties preventing informed consent.[34] Toddlers were invited to the Centre for the Developing Brain, St Thomas' Hospital, London, for neurodevelopmental assessment at 18 months post expected delivery date; appointments were made according to family availability as close as possible to this time point. Inclusion criteria for our follow-up study were: mother and baby attendance for MRI at term-equivalent age; completed toddler neurodevelopmental assessment. These inclusion criteria were met by 509 toddlers by the date of closure for this analysis (26 February 2020). Of the 509 toddlers, 51 were one of a twin pregnancy, and three were one of a triplet pregnancy; the sample contained 22 sibling pairs and one set of triplets.

### Maternal variables
Maternal age, parity, body mass index (BMI), ethnicity and postcode were collected at enrolment into the DHCP study. Our sample was ethnically representative of the surrounding geographical area. Parity was coded as 0, 1, 2 or ≥3 previous children. Index of Multiple Deprivation (IMD) rank was computed from the current maternal postcode using the 2019 IMD classification; it combines locality specific information about income, employment, education, health, crime, housing and living environment, thus providing a proxy for family socioeconomic status.[35] Lower IMD rank corresponds to greater social deprivation. Our sample was generally less deprived than

the surrounding geographical areas, as well as the UK as a whole, reflecting trends observed in other UK longitudinal studies.[36]

*Maternal depressive symptoms* were measured using the Edinburgh Postnatal Depression Scale (EPDS)[37] on the day of infant's MRI at term-equivalent age. Mothers of infants born at term were tested in the first few weeks postnatally, whereas mothers of preterm-born infants were tested once they reached term-corrected age. The EPDS is a 10-item screening questionnaire completed by mothers, with higher scores reflecting a higher likelihood of depressive disorders. A score of 13 can be used as a cut-off, indicating high-level symptoms, although a cut-off of 11 maximises the sensitivity and specificity of the screening tool for depression.[38] Mothers completed the EPDS independently in a private room in our centre, with no interaction with the researcher. Participants were informed that the results would be discussed with them and consented to information being shared with their general practitioner in the case of high scores. The EPDS questionnaire was scored by a member of the DHCP team.[34]

### Child variables

Infant *clinical characteristics* were gathered from clinical notes where available, or from maternal report, and included: sex, gestational age at birth, birth weight and pregnancy size (singleton/twin/triplet).

*Behavioural outcomes* were assessed using the Child Behaviour Checklist/$1^{1/2}$–5 (CBCL), a parent-completed 100-item questionnaire, in which the parent rates the child's behaviour over the preceding 2 months using a 3-point Likert scale ('not true', 'somewhat or sometimes true' and 'very true or often true'). Responses are categorised into syndrome profiles, and these are subsequently grouped into internalising (emotional reactivity, anxiety/depression, somatic complaints and withdrawal), externalising (attention problems, aggressive behaviour) and total (internalising, externalising, sleep and other) problem scales. Higher scores indicate increased emotional and behavioural problems. Total scores are classified into a normal range (<83rd centile, T<60), borderline range (83rd–90th centile, T 60–63) and clinical range (>90th centile, T≥64).[39] The CBCL is known to have high reliability, validity and cross-informant agreement for measuring children's emotional and behavioural problems.[39]

We used the Quantitative Checklist for Autism in Toddlers (Q-CHAT) as an additional behavioural screening tool to broaden the exploration of mental health outcomes in toddlers. The Q-CHAT is a parent-completed 25-item questionnaire, in which the child's behaviour is scored on a 5-point (0–4) frequency scale. Higher total scores correspond to a higher frequency of behaviours also observed in autism spectrum conditions. The Q-CHAT shows good test–retest reliability, face validity and specificity, yet poor positive predictive value for autism,[40 41] highlighting that higher Q-CHAT scores may reflect developmental immaturity rather than autism.[41]

*Cognitive assessment* was performed using the Bayley Scales of Infant and Toddler Development–Third Edition (Bayley-III). The Bayley-III provides scores for a child's overall cognitive, language and motor development. The cognitive standardised composite score was used in this study; scores between 70 and 84 indicate mild cognitive impairment, scores between 55 and 69 indicate moderate impairment and scores lower than 55 indicate severe impairment.[42] Reliability and validity of the Bayley-III have been shown to be robust,[43] although some studies report its underestimation of developmental problems.[44]

Assessments were carried out by staff experienced in the neurocognitive assessments of toddlers.

### Analysis

Descriptive statistics and one-way Analysis of Variance tests were performed in IBM SPSS Statistics for Windows V.25. All other analyses were carried out in Stata V.16.

Multiple imputation (MI) was carried out to account for missing data in CBCL (11/509, 2.24%), Q-CHAT (9/509, 1.8%), maternal EPDS (73/509, 14.3%), maternal BMI (27/509, 5.3%) and IMD rank (3/509, 0.6%). Variables were imputed simultaneously using the 'mi impute chained' procedure that performs imputation by chained equations. The imputation models had the same structural form as the analysis models and included all variables that appear in the corresponding analysis models (maternal EPDS, maternal BMI, multiple pregnancy, parity, IMD rank, gestational age at birth, birth weight, sex, corrected age at assessment and Bayley III Cognitive Composite score). In the imputation models, we also included variables that were associated with the incomplete variables at the 20% level. As such, maternal age was included in the imputation model because it was found to be a significant predictor of the total CBCL raw score (p=0.001), the Q-CHAT score (p=0.021) and EPDS score (p=0.122) when it was included as an independent variable in regression models.

Maternal EPDS and CBCL were imputed using Poisson regression; Q-CHAT, maternal BMI and the IMD rank were imputed using linear regression. Fourty MI data sets were created. To assess the stability of our MI parameters, we extracted the Monte Carlo error of each parameter estimate and examined whether the error for the coefficient was less than 10% of the parameter's SE estimate. MI estimates were used for the primary analyses and compared with the estimates from complete-case (CC, individuals who had no missing data preimputation) analyses. Conditional normality was inspected in the CC analyses using QQ plots of the residuals of the models. Sensitivity analyses with and without extreme values were conducted. Initially, we fit the model using all available data, constructed the residuals and examined the QQ plot. Extreme values were then removed, models refitted without these values, and new QQ plots of residuals constructed again

to check for any new extreme values. This process was repeated as many times as needed to remove all extreme values. During this process, the resulting estimates from the models were being examined as to whether they had substantially changed. We found that the removal of extreme values did not make any difference to the estimated parameters, and hence present the results from the full sample.

The analysis models were multiple linear regressions fitted using the 'mi estimate' procedure, which estimates effects after application of Rubin's rules.[45] To account for the small amount of clustering in our data (twin/triplet siblings), the models' SEs were obtained using Stata's robust cluster estimator 'vce(cluster *idvar*)'. For continuous variables, Cohen's $f^2$ effect sizes were calculated using $f^2 = \left(R^2_{AB} - R^2_A\right) / \left(1 - R^2_{AB}\right)$, where $R^2_{AB}$ is the $R^2$ value from a regression model that includes the variable of interest as well as all the covariates used in the model, and $R^2_A$ is the $R^2$ value from the regression model that includes only the covariates.[46 47] For binary variables, Cohen's $f^2$ effect sizes were produced after estimating first the Cohen's d using the formula: $f^2 = \frac{d}{2k}$, where k is the number of groups. As a measure of dispersion, Cohen's d used the average root mean-square error over the MI data sets. Adjusted $R^2$ values after MI were extracted after estimating the model with the user-written 'mibeta' command with the 'fisherz' option,[48] which calculates $R^2$ measures for linear regression with MI data. The significance of the joint effect of the categorical variable parity was assessed using 'mi test', which performs Wald tests of composite linear hypotheses.

*Primary outcome measures* were children's total CBCL raw score and Q-CHAT score. S*econdary outcome measures* were CBCL internalising and externalising scores. The effect of maternal EPDS score was adjusted for IMD rank, maternal age, maternal BMI, maternal parity, pregnancy size and the following child variables: continuous gestational age, birth weight, Bayley-III cognitive composite score and corrected age at assessment. The interaction between PTB and maternal depressive symptoms was explored using a CC analysis in both CBCL and Q-CHAT models, using a dichotomised measure of gestational age. EPDS, CBCL and Q-CHAT scores were compared between term (≥37 weeks gestation) and preterm infants (<37 weeks gestation) using the CC data set. Our regressions were, thus, run two times: with and without the interaction term.

As all mothers had their EPDS score measured near term (or term-corrected in the case of mothers of preterm infants), we further investigated the association between time elapsing between baby's birth and mother's EPDS assessment and EPDS score, in order to avoid erroneously identifying 'baby blues' in mothers of term-born infants versus postnatal depression in mothers of preterm infants. This post hoc analysis was performed using Poisson regression.

### Patient and public involvement

The current study was developed in consultation with the Weston Programme for Family Centered Research, which involves parents to define what research is valuable to them and to allow them to lead it with support from the scientists in the Centre for the Developing Brain.

## RESULTS

### Descriptive statistics

Our sample of 509 toddlers was followed up at a median-corrected age of 18.4 months (range 17.3–24.3 months). 51 (10.0%) of these were twins, and 3 (0.59%) were triplets. Of the 509, 21 (4.13%) mothers scored above a clinical cut-off (≥13) on the EPDS[37 38]; the distribution of maternal EPDS scores is shown graphically in online supplemental figure 1. Demographic data are shown in table 1. Complete data were available for 400 (78.6%) participants. Missing data were imputed and thus all 509 subjects were included in the primary and secondary analyses. One participant was excluded from the cognition analysis after examining the quintiles of the residuals against the theoretical quintiles of a normal distribution. The mean CBCL T score was 46.9 (SD 9.5) (table 1); using CBCL-specified cut-offs,[39] 449 (90.2%) of participants had a CBCL score in the normal range, 30 (6.0%) were borderline and 19 (3.8%) scored in the clinical range. The mean Q-CHAT score was 30.5 (SD 9.3) (table 1). The mean Bayley III Cognitive Composite score in our sample was 100 (SD 11.4) (table 1), which corresponds to the standardised test mean[42]; 480 (94.3%) of participants had a normal cognitive score, 24 (4.7%) had mild impairment, 5 (1%) had moderate impairment and nil had severe impairment. This distribution is not dissimilar from that of the normative sample.[42]

### Association between maternal EPDS score and toddler CBCL and Q-CHAT scores

Predictors of children's CBCL and Q-CHAT scores after MI are shown in table 2. Higher maternal EPDS score was associated with children's higher CBCL total score (B=0.93, 95% CI 0.43 to 1.44, p<0.001, $f^2$=0.05) and Q-CHAT score (B=0.27, 95% CI 0.03 to 0.52, p=0.031, $f^2$=0.01) (table 2). These associations are presented graphically in figure 1 and figure 2, respectively. Boys had higher CBCL and Q-CHAT scores than girls. Higher Q-CHAT scores were associated with lower IMD rank (ie, greater socioeconomic deprivation) and lower Bayley-III cognitive composite scores. Parity was not a significant predictor of outcome in any of the models (table 2).

Maternal EPDS score did not disproportionately affect preterm children with respect to CBCL or Q-CHAT scores (table 3).

### Association between maternal EPDS score and toddler CBCL internalising and externalising scores

Higher maternal EPDS score was associated with both internalising (B=0.22, 95% CI 0.08 to 0.36, p<0.01,

**Table 1** Sociodemographic, maternal and clinical characteristics (n=509).

| Variable | Number (%)* |
|---|---|
| Sex: male | 274 (53.8) |
| Index of multiple deprivation (IMD) quintiles | |
| 1 (least deprived) † | 65 (12.8) |
| 2 | 87 (17.2) |
| 3 | 108 (21.3) |
| 4 | 173 (34.2) |
| 5 (most deprived) | 73 (14.4) |
| Gestational age at birth (weeks), median (range) | 39.7(20–43) |
| Gestational category | |
| Extremely preterm (<28 weeks) | 18 (3.5) |
| Very preterm (28–32 weeks) | 28 (5.5) |
| Late preterm (32–37 weeks) | 51 (10.0) |
| Term (≥37 weeks) | 412 (80.9) |
| Birth weight (g), median (range) | 3290 (450–4750) |
| Multiple pregnancy | 54 (10.6) |
| Maternal parity | |
| 0 | 332 (65.2) |
| 1 | 124 (24.4) |
| 2 | 32 (6.3) |
| 3+ | 21 (4.2) |
| Maternal BMI (kg/m$^2$), median (range) | 23.2 (15.3–43.6) |
| Maternal age at infant's birth (years), mean (SD) (range) | 34.2 (4.8) (17–52) |
| Maternal ethnicity | |
| White | 272 (53.4) |
| Black/Black British | 56 (11.0) |
| Asian/Asian British | 28 (5.5) |
| Chinese | 18 (3.5) |
| MixedWhite and Asian | 4 (0.8) |
| MixedWhite and Black | 4 (0.8) |
| Any other | 30 (5.9) |
| Do not wish to answer | 9 (1.8) |
| No data | 88 (17.3) |
| Bayley III cognitive composite score, mean (SD) (range) | 100 (11.4) (55–125) |
| CBCL total T score, mean (SD) (range) | 46.9 (9.5) (28–69) |
| Q-CHAT total score, mean (SD) (range) | 30.5 (9.3) (8–70) |
| EPDS score, median (range) | 4 (0–28) |
| EPDS score, n (%) | |
| <13 | 415 (8.2) |
| ≥13 | 21 (4.1) |

Continued

**Table 1** Continued

| Variable | Number (%)* |
|---|---|
| No data | 73 (14.3) |

*Unless otherwise specified.
†Quintile 1 corresponds to the highest, least deprived, IMD rankings.

$f^2$=0.03) and externalising (B=0.40, 95% CI 0.20 to 0.61, p<0.001, $f^2$=0.05) symptoms in children (online supplemental table 1 and 2), respectively). Comparison of the imputed model analyses to the CC analyses showed that results were consistent for the CBCL model (online supplemental table 3). Comparison for the Q-CHAT model showed that maternal EPDS was a significant predictor in the imputed model, but not in the CC analysis (online supplemental table 3).

### Effect of time-lag between baby's birth and mother's EPDS assessment and EPDS score

Mothers who gave birth prematurely (<37 weeks gestation) had their EPDS score assessed on average 7.7 weeks later post delivery than mothers who gave birth at term (preterm participants M=8.9 (SD 4.8), term participants M=1.2 (SD 1.3); t(99.4)=15.5, p<0.001). The time-lag between birth and EPDS assessment did not predict maternal EPDS score, and there was no evidence of a significant interaction between gestation and birth-to-assessment time-lag (online supplemental tables 4 and 5, respectively).

## DISCUSSION
### Principal findings

Our results showed that more maternal self-reported depressive symptoms shortly after birth were associated with greater parent-reported toddlers' behavioural problems. Given that fewer than 5% of the mothers in our cohort had EPDS scores above a clinical threshold,[37] our findings indicate that even subclinical depressive symptoms—that is, not only diagnostic postnatal depression—adversely impact children's behavioural outcomes. In addition, our cohort was typically developing with few CBCL scores reaching a concerning threshold; our results could be interpreted within the conceptual framework of mental illness lying on a continuum with typical behavioural traits.[49] Our findings further showed that PTB did not influence the association between self-reported maternal depressive symptoms and parent-reported infants' behavioural outcomes in toddlerhood. This indicates that in this context, PTB may not be regarded as a vulnerability or plasticity factor. Interestingly, mothers of preterm infants did not report more depressive symptoms compared with mothers of term infants in this study.

**Table 2** CBCL and Q-CHAT model predictors using multiple imputation without interaction (Cf. (online supplemental table 3) for complete-case analysis)

|  | CBCL | | | QCHAT | | |
|---|---|---|---|---|---|---|
|  | B (95% CI) | p | f² | B (95%CI) | p | f² |
| Maternal EPDS | 0.93 (0.43 to 1.44) | <0.001 *** | 0.05 | 0.27 (0.03 to 0.52) | 0.031* | 0.01 |
| Maternal BMI | −0.09 (−0.44 to 0.26) | 0.621 | – | 0.06 (-0.13 to 0.24) | 0.538 | – |
| Multiple pregnancy | 3.15 (-3.07 to 9.37) | 0.320 | – | 1.33 (-2.62 to 5.28) | 0.509 | – |
| Parity |  |  |  |  |  |  |
| 1 | −2.52 (−5.96 to 0.93) | 0.151 | – | −2.14 (−4.02 to -0.27) | 0.025 † | – |
| 2 | −3.23 (−9.16 to 2.70) | 0.285 | – | 0.88 (-1.99 to 3.75) | 0.548 | – |
| 3+ | −1.37 (−8.36 to 5.61) | 0.699 | – | −0.49 (−4.57 to 3.60) | 0.815 | – |
| IMD rank | −1.48 (−3.33 to 0.37) | 0.117 | – | −1.50 (−2.60 to -0.40) | 0.008 ** | 0.02 |
| Gestational age at birth (weeks) | 0.10 (-0.65 to 0.85) | 0.786 | – | 0.26 (-0.17 to 0.70) | 0.233 | – |
| Birth weight (kg) | 0.56 (-2.65 to 3.78) | 0.731 | – | −1.24 (−2.93 to 0.46) | 0.151 | – |
| Sex:female | −4.14 (−6.96 to -1.31) | 0.004 ** | 0.06 | −1.95 (−3.42 to -0.48) | 0.009 ** | 0.05 |
| Corrected age at assessment (months) | −0.90 (−2.17 to 0.37) | 0.166 | – | −0.16 (−0.91 to 0.59) | 0.677 | – |
| Cognition | −0.05 (−0.20 to 0.09) | 0.467 | – | −0.27 (−0.35 to -0.20) | <0.001 *** | 0.12 |

CBCL model adjusted $R^2$=0.0676. Q-CHAT model adjusted $R^2$=0.193.
Effect size (Cohen's $f^2$, calculated from squared part correlations for predictors significant to 0.05): 0.02=small, 0.15=medium and 0.35=large0.[46]
– Indicates data not given, as predictor not significant to 0.05.
**p<0.05; **p<0.01; ***p<0.001.
†Wald test of whole parity variable in Q-CHAT model: $F_{(3, 476.9)}$ = 1.88, p=0.133.
B, unstandardised coefficient; CBCL, Child Behaviour Checklist score at 18 months; Cognition, infant Bayley III score at 18 months; Maternal EPDS, maternal Edinburgh Postnatal Depression Scale score at term-equivalent age; Corrected age at assessment (months), age at behavioural assessment, corrected for gestational age; Parity, dummy variable, one/two/three+ previous child(ren); Multiple pregnancy, dummy variable of twin/triplet pregnancy; Q-CHAT, Quantitative Checklist for Autism in Toddlers score at 18 months.

### Comparison to prior literature

Our results with respect to internalising and externalising symptoms are in line with previous studies, including large population cohort studies, which showed an association between postnatal maternal depression and young children's emotional and behavioural problems.[11] Another previous study in 18-month old toddlers found that maternal depression was associated with internalising and dysregulated behaviour, but not externalising symptoms.[50] This difference between our and Conroy

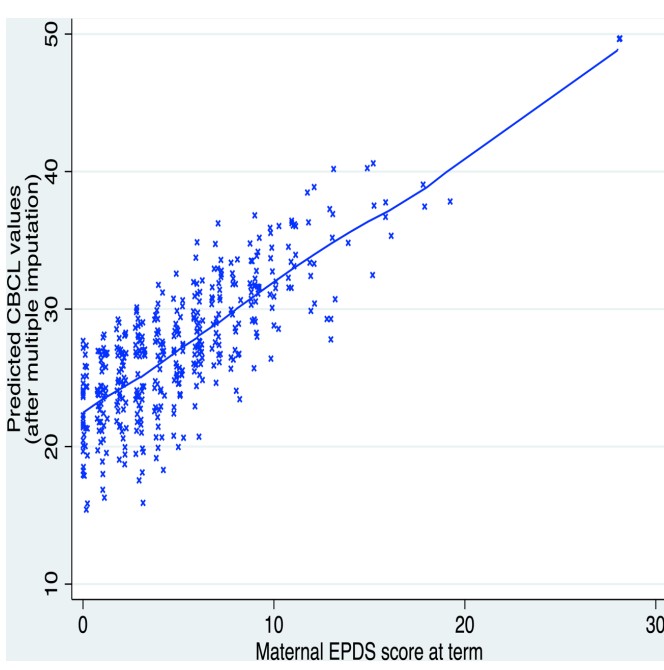

**Figure 1** Children's predicted CBCL scores at 18 months are positively correlated to the maternal EPDS score at term-equivalent age. CBCL, Child Behaviour Checklist score; EPDS, Edinburgh Postnatal Depression Scale.

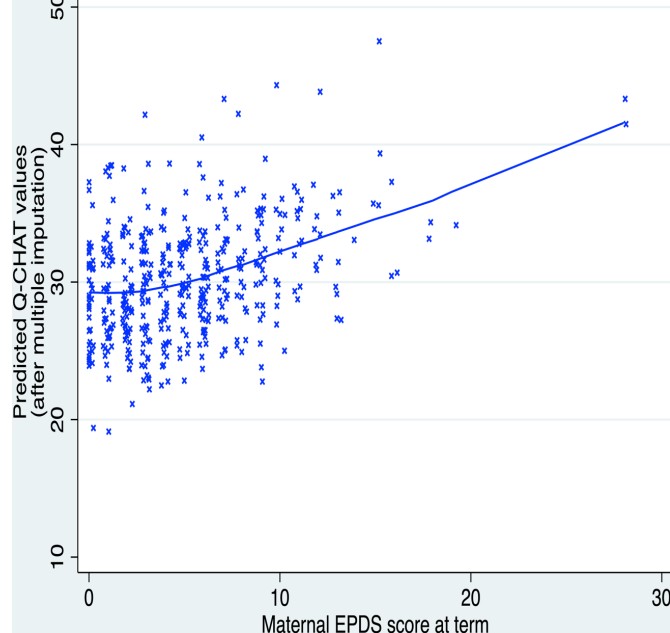

**Figure 2** Children's predicted Q-CHAT scores at 18 months are positively correlated to the maternal EPDS score at term-equivalent age. EPDS, Edinburgh Postnatal Depression Scale; Q-CHAT, Quantitative Checklist for Autism in Toddlers.

**Table 3** CBCL and Q-CHAT model predictors using complete-case analysis with interaction of 'EPDS x Term'

| | CBCL | | Q-CHAT | |
|---|---|---|---|---|
| | B (95% CI) | p | B (95% CI) | p |
| Maternal EPDS | 0.89 (−0.24 to 2.02) | 0.121 | 0.24 (−0.28 to 0.75) | 0.365 |
| Maternal BMI | −0.01 (−0.39 to 0.37) | 0.955 | 0.00 (−0.16 to 0.17) | 0.982 |
| Multiple pregnancy | 1.76 (−6.65 to 10.17) | 0.681 | 0.97 (−2.07 to 4.01) | 0.532 |
| Parity | | | | |
| 1 | −2.75 (−6.49 to 0.99) | 0.149 | −1.42 (−3.30 to 0.46) | 0.139 |
| 2 | −3.49 (−10.36 to 3.37) | 0.317 | 0.16 (−2.84 to 3.16) | 0.917 |
| 3+ | −1.17 (−9.69 to 7.35) | 0.788 | −1.13 (−4.24 to 1.98) | 0.476 |
| IMD rank | −1.41 (−3.54 to 0.73) | 0.195 | −1.68 (−2.64 to -0.72) | 0.001 ** |
| Gestation: term | 1.25 (−8.34 to 10.85) | 0.797 | 2.64 (−1.74 to 7.02) | 0.236 |
| Birth weight (kg) | −1.01 (−4.08 to 2.05) | 0.516 | −2.25 (−3.73 to -0.78) | 0.003 ** |
| Sex: female | −4.64 (−7.83 to -1.44) | 0.005 ** | −2.22 (−3.72 to -0.71) | 0.004 ** |
| Corrected age at assessment (months) | −0.83 (−2.27 to 0.62) | 0.261 | −0.39 (−1.18 to 0.04) | 0.335 |
| Cognition | −0.03 (−0.20 to 0.14) | 0.720 | −0.22 (−0.29 to −0.15) | <0.001 *** |
| EPDS x Gestation:Term | −0.01 (−1.30 to 1.28) | 0.991 | −0.02 (−0.60 to 0.56) | 0.950 |

*p<0.05; **p<0.01; ***p<0.001.
CBCL model adjusted $R^2$=0.0865. Q-CHAT model adjusted $R^2$=0.215.
B, unstandardised coefficient; CBCL, Child Behaviour Checklist score at 18 months; Cognition, infant Bayley III score at 18 months; Maternal EPDS, maternal Edinburgh Postnatal Depression Scale score at term-equivalent age; EPDS x Gestation:Term, interaction term between maternal EPDS score and term gestation at birth; Corrected age at assessment (months), age at behavioural assessment, corrected for gestational age; Parity, dummy variable, one/two/three+ previous child(ren); Multiple pregnancy, dummy variable of twin/triplet pregnancy; Q-CHAT, Quantitative Checklist for Autism in Toddlers score at 18 months; Gestation: term, dummy variable, term (≥37 weeks) versus preterm (<37 weeks) gestation at birth.

*et al*'s findings may have arisen from their exclusion of infants born <36 weeks and their use of a clinical diagnosis of depression for mothers, rather than the continuous self-reported approach we employed. Interestingly, our finding that even subclinical depressive symptoms may adversely impact parent-reported child behavioural outcomes is in line with recent data showing that low-level as well as high-level depressive symptoms are associated with internalising and externalising symptoms in children aged 3 years.[51]

The results showing an association between maternal postnatal depressive symptoms and the Q-CHAT are less robust and need to be interpreted with caution. First, these results must be viewed in the context of the Q-CHAT having a low positive predictive value for autism, with the measure perhaps being more reflective of developmental immaturity.[41] Although some prior studies have shown an association between antenatal maternal depression and offspring's ASD,[10 52] and postnatal depression has been suggested as a potential focus of cross-domain studies of ASD,[53] there is no clear aetiological role of maternal postnatal depression in the development of ASD per se. Also, given that mothers with ASD are more likely to suffer from perinatal depression than mothers without ASD,[54] and ASD is highly heritable,[55] maternal depression may actually be a confounding rather than causative factor in our observed results. Overall, therefore, our findings with respect to the Q-CHAT do not provide support for a role

of maternal depression in the aetiology of autism traits, but rather suggest that maternal depression can influence toddler behaviour.

The finding that preterm infants were not disproportionately affected by maternal depressive symptom supports. Hadfield *et al*'s findings that maternal distress at 9 months did not differentially impact very preterm (<34 weeks) or late preterm (34–36$^{+6}$ weeks) infants with respect to socioemotional outcomes, although paternal distress did have an impact on very preterm infants' outcomes.[24] However, our results differ from Gueron-Sela *et al*'s finding that very preterm (28–33 weeks) 12-month old infants' social outcomes were more influenced by maternal emotional distress at 6 months than term infants' outcomes.[23] The inconsistent findings may be due to methodological differences: for instance, our infant assessment being conducted at 18 months corrected age when social competency is more developed, our assessment of maternal depressive symptoms being in the very early postnatal period, or our use of the CBCL and Q-CHAT tools as markers of toddler behaviour. Importantly, the lack of support for a diathesis–stress or differential susceptibility model of maternal mental state on preterm infants in our study must be viewed in the context of our results also showing no difference in CBCL and Q-CHAT scores between term and preterm infants. This is in contrast to the existing literature that preterm infants are more likely than term infants to develop

behavioural problems, such as ADHD, in childhood and adolescence.[20 33] It is possible that the phenotypes of neurodevelopmental and neuropsychiatric disorders assessed with the chosen behavioural measures may not be sufficiently expressed at 18 months corrected age.[56] In addition, as briefly discussed above, much of the existing literature emphasises the risk of extreme (<28 weeks) or very preterm (28–33 weeks) birth on later behavioural outcomes,[20 33] whereas only 3.5% and 5.5% of our participants fell within the extreme and very preterm group, respectively, and we, thus, may not have the power to show any subtle effects.

### Strengths and limitations of the study

The strengths of this study lie primarily in its large sample and prospective data collection. Moreover, the use of MI methodology has facilitated retention of a complete dataset, thus minimising non-response bias and increasing parameter precision. A strength in comparison to prior population cohort studies is that we assessed very early maternal depressive symptoms, and our sample is perhaps more representative of today's society—with increasing maternal age—than large cohort studies conducted in the 1990s–2000s. Given the complex interplay of biological and environmental factors in the aetiology of behavioural disorders, the inclusion of a substantive proportion of preterm infants in our cohort also offers an important insight into the role of PTB in behavioural outcomes; moreover, our results represent the full gestational spectrum, rather than discrete gestational categories. In addition, using maternal depressive symptoms as a continuous, rather than dichotomous, variable allows a more nuanced understanding of the role maternal postnatal depressive symptoms may play in influencing children's outcomes.

There are several limitations to this study that necessitate our findings to be considered with caution. First, differences in birth-to-EPDS-assessment time lags are a potential confounder, given the time-sensitive nature of early-onset temporary baby blues and late-onset pathological postnatal depression. Mothers of infants born at term were assessed early postdelivery, within the period one would anticipate baby blues to present, whereas mothers of preterm participants were on average assessed later, when postnatal depression predominates.[1 57] Although our post hoc analyses showed that the time elapsed from birth to EPDS assessment was not associated with maternal EPDS score, providing reassurance that our assessments of mothers of term-born infants were not inflated by the common temporary symptoms of baby blues, it is possible that we did not capture the full extent of late-onset depressive symptoms in mothers of term-born infants. This may explain why maternal EPDS scores did not differ between preterm and term groups in our complete data set analysis, contrary to the current literature,[31] as well as why our rate of postpartum depression, using an EPDS cutoff of 13, was low (4.1%) compared with the previously documented UK community prevalence rate of 8.9% at

8 weeks postpartum.[58] Our results must, therefore, be interpreted with some caution.

Second, although statistical techniques were used to impute missing data and mitigate this problem, 14.3% of maternal EPDS scores were missing. This rate of missingness may relate to some mothers being reluctant to complete a questionnaire at the time their child is having an MRI or due to simultaneous childcare duties. Third, a number of important confounders that are likely to affect children's behavioural outcomes were not assessed in this study, including genetic risk for psychiatric disorders,[59] parental psychiatric comorbidities,[50] chronicity of postnatal depressive symptoms,[51] antenatal maternal depression, paternal depression, subsequent parent–infant attachment, and interparental conflict.[11] Thus, we are unable to conclude whether our observed associations between early postnatal maternal depressive symptoms and children's behavioural outcomes are moderated or mediated by other parental and/or psychiatric factors.

Fourth, while our study included a reasonable proportion of preterm infants (97/509, 19%), our sample was not random, as preterm children were selectively recruited for the DHCP; indeed, preterm infants are over-represented in our sample when compared with the UK population incidence (7.3%),[60] which may limit the study's generalisability to the general population. This over-representation of preterm infants may explain why our mean maternal age is higher than the national mean age of 30.7,[61] given that increasing maternal age is associated with increased risk of adverse pregnancy outcomes.[62] Our observed large maternal age range in itself also poses a limitation on the generalisability of our findings to the general population, and further research would be necessary to identify a possible moderation effect of high maternal age on both EPDS scores and child behavioural outcomes. Furthermore, although a 19% prevalence of PTB is high for a community sample, the proportion of very and extreme preterm infants in our sample is small, and this may not have provided sufficient power to detect any differential susceptibility effect of PTB on outcomes.

Sixth, the effect sizes of the association between maternal EPDS score and behavioural problems were small; this raises questions regarding the clinical significance of our findings and potentially explains some of the inconsistency between this and previous studies. Even within our analyses, the association between maternal depressive symptoms and Q-CHAT scores was not observed in our CC analysis, thus calling into question the validity of this result. It is also important to highlight again the poor positive predictive value of the Q-CHAT for autism[41]; higher Q-CHAT scores do not imply a diagnosis of ASD, and this distinction may also explain the contrast to previous studies.

Finally, it is well documented that maternal depression influences reporting of Q-CHAT[63] and CBCL scores.[64] Our study used maternal report of maternal depressive symptoms, and our outcome measures were parent-completed questionnaires; despite the CBCL showing

good cross-informant agreement,[39] it is, thus, possible that reporting bias with common method variance could have skewed our results.

## Implications of our findings

Of greatest importance to clinicians and policymakers is our finding that even *subclinical* self-reported maternal depressive symptoms are associated with parent-reported behavioural outcomes of offspring. This has significant implications for the risk stratification of women and their babies in the postnatal period, during which contact with medical professionals is already established. Identifying high-risk families and providing appropriate supportive measures at the early postnatal stage may help to prevent future psychiatric morbidity.

## Future research

Further follow-up of large cohorts of preterm and term infants, to an age when behavioural phenotypes may become more pronounced, is needed to investigate whether the long-term developmental trajectories of term and ex-preterm infants are differentially susceptible to changes of postnatal maternal mental health. Future research should consider both maternal and paternal mental health as well as socioeconomic and environmental factors on child outcomes. Such follow-up should use independent, objective assessments of child behavioural outcomes in order to avoid the common method variance inherent to parent-reported measures. Finally, it is crucial for future research to elucidate the interplay of biochemical and neurodevelopmental changes that may mediate and confound the translation of environmental exposures into distal behavioural phenotypes.

## CONCLUSION

This prospective longitudinal cohort study found no evidence to support the concept of PTB as a vulnerability or plasticity factor with respect to the effect of maternal depressive symptoms on behavioural development. However, we showed that early subclinical maternal postnatal depressive symptoms were associated with behavioural problems in children on parent-reported measures. This adds to the increasing body of literature indicating the role of subclinical and early postnatal depressive symptoms in the aetiology of childhood behavioural disorders. These findings are of great relevance to child and public health, and further research may strengthen its implications for developing strategies to facilitate effective screening and support for women and children, enabling all to reach their full potential.

## Author affiliations
[1]Centre for the Developing Brain, School of Bioengineering and Imaging Sciences, Faculty of Life Sciences & Medicine, King's College London, London, UK
[2]Department of Biostatistics and Health Informatics, Institute of Psychiatry, Psychology and Neuroscience, King's College London, London, UK
[3]Forensic and Neurodevelopmental Sciences, Institute of Psychiatry, Psychology and Neuroscience, King's College London, London, UK
[4]Department of Child and Adolescent Psychiatry, Institute of Psychiatry, Psychology and Neuroscience, King's College London, London, UK

**Contributors** We thank all DHCP investigators for their contribution to the study. We thank Dr Oliver Gale-Grant MRes (Centre for the Developing Brain, King's College London; Department of Forensic & Neurodevelopmental Sciences, King's College London) for providing the IMD rank data. We are very grateful to the families who generously took part in this research. Conceptualisation: SC, DE, CN. Methodology: IK, GV, SC, AP, DE, CN. Investigation: SF, AC. Data curation: IK, AL. Formal analysis: IK, GV, AP, CN. Writing—original draft preparation: IK. Writing—review and editing: IK, GV, AL, SF, AC, SC, AP, DE, CN. Visualisation: IK, GV. Funding acquisition: SC, DE. Supervision: AP, DE, CN. Guarantor: CN.

**Funding** The DHCP project was funded by the European Research Council under the European Union Seventh Framework Programme (FR/2007-2013)/ERC Grant Agreement number 319456. The authors acknowledge infrastructure support from the National Institute for Health Research Mental Health Biomedical Research Centre at South London, Maudsley NHS Foundation Trust, King's College London, the National Institute for Health Research Mental Health Biomedical Research Centre at Guys, and St Thomas' Hospitals NHS Foundation Trust. The study was also supported in part by the Engineering and Physical Sciences Research Council/Wellcome Trust Centre for Medical Engineering at King's College London (grant WT 203148/Z/16/Z) and the Medical Research Council (UK) (grants MR/K006355/1 and MR/L011530/1) and the MRC Centre for Neurodevelopmental Disorders at King's College London. AP receives a NIHR SI award (NF-SI-0617-10120). AL is supported by the UK Medical Research Council (MR/N013700) and King's College London member of the MRC Doctoral Training Partnership in Biomedical Sciences. The views expressed are those of the authors and not necessarily those of the NHS, the NIHR, or the Department of Health and Social Care.

**Competing interests** ADE received financial support from the EU-AIMS-Trials (European Research Council under the European Union Seventh Framework Programme) as co-principal investigator. ADE received consulting fees from Chiesi Farmaceutici (advice on neuroprotection in newborn infants) and Medtronix (unpaid participation in Scientific Advice Committee). ADE has a patent on Xenon as an organ protectant (Number P023708WO). ADE was Chair of the Data Monitoring and Ethics Committee for the Baby-Oscar Trial, and served on the Data Monitoring and Ethics Committee for the PAEN Trial. There are no other relationships or activities that could appear to have influenced the submitted work.

**Patient and public involvement** Patients and/or the public were involved in the design, or conduct, or reporting, or dissemination plans of this research. Refer to the Methods section for further details.

**Patient consent for publication** Not applicable.

**Ethics approval** This study involves human participants and was approved by UK National Research Ethics Authority (14/LO/1169) and conducted in accordance with the World Medical Association's Code of Ethics (Declaration of Helsinki). Written informed consent was given by children's carer(s) at recruitment into the study.

**Provenance and peer review** Not commissioned; externally peer reviewed.

**Data availability statement** Data are available upon reasonable request.

**ORCID iD**
Ira Kleine http://orcid.org/0000-0002-6500-9407

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
