## [Reviewer comments · BMJ Open]

ARTICLE DETAILS

TITLE (PROVISIONAL)	Postnatal maternal depressive symptoms and behavioural outcomes in term- and preterm-born toddlers: a longitudinal UK community cohort study.
AUTHORS	Kleine, Ira; Vamvakas, George; Lautarescu, Alexandra; Falconer, Shona; Chew, Andrew; Counsell, Serena; Pickles, Andrew; Edwards, David; Nosarti, Chiara

VERSION 1 – REVIEW

REVIEWER	Lin, Betty University at Albany
REVIEW RETURNED	08-Dec-2021

GENERAL COMMENTS	The present analyses examined links between maternal postnatal depressive symptoms and toddler adjustment outcomes, and whether preterm birth would heighten susceptibility to maternal postnatal depressive symptoms for better and for worse. There were many study strengths, including the large sample size, longitudinal/ birth cohort design, and use of multiple imputation to address missing data. Additionally, it is readily apparent that the authors were thorough in testing potential alternate explanations (e.g., such as evidenced by testing the effects of time lag of EPDS assessment on EPDS scores). The author also took strides to be transparent about their analytic approach and available data (e.g., such as evidenced by comparisons of analyses using MI and using only complete data). I had some questions and concerns, which I detail below. - From a conceptual standpoint, my understanding is that the differential susceptibility hypothesis does not regard preterm birth as a susceptibility factor. Instead, there is some evidence that indices of fetal growth (especially low birthweight and intrauterine growth restriction) may represent proxies suggestive of susceptibility (see Pluess & Belsky, 2011, Development & Psychopathology; Seckl, 2001, Molecular & Cellular Endocrinology). Accordingly, I think it's fine to frame preterm birth as a proxy variable, but would adjust language here for clarity and discuss implications accordingly (i.e., insofar as it is a proxy variable, it is an imperfect capture of putative susceptibility per differential susceptibility).- Discussion of background literature in the introduction regarding differential susceptibility in relation to adjustment outcomes felt somewhat overly cursory, particularly regarding potential differential effects of postnatal depressive symptoms on emotional and behavioral adjustment, ASD symptoms, and cognitive development. Related, please note that the relations of postnatal depressive symptoms with cognitive development is not currently included in the stated aims.
--

	 - Considering the centrality of preterm birth for the conceptual model, it wasn't clear to me whether gestational age – which was entered in the analyses to reflect preterm birth – was coded as a continuous (gestational age) or dichotomous (preterm or term birth) variable. While I'm generally a proponent of retaining continuous properties of variables for analyses, if the condition of preterm v. term birth is as foundational as the authors suggest given the conceptual framework, then the reliance on the continuous variable v. dichotomizing the variable should be justified. - Somewhat related to the above, there were several places throughout the manuscript where the authors were inconsistent with term usage, which made it difficult to distinguish whether terms were intentionally changed (e.g., term birth & gestational age; maternal depression & maternal depressive symptoms) or used interchangeably. For clarity, it would help if the authors used the same terms to describe the same constructs and to additionally consider explicit statements differentiating variations of terms that are salient. Similarly, it would help to clarify if "term corrected" only applied to infants who were born preterm (i.e., before 37 weeks). Finally, I also found it confusing that gestational age was generally viewed as a variable of primary interest but was sometimes described as a "confounder" (e.g., see p. , line 37). - Also related to the issue of preterm birth as a proxy variable, I don't understand the rationale for also include birth weight as a covariate in analyses, especially because the latter may represent a better proxy of susceptibility than preterm birth. What were correlations between preterm birth, gestational age, and birth weight? I'm wondering if there may have been issues with multicollinearity. It would help to include correlations among all covariates and key study variables in general, including only one or the other in models, and potentially additionally investigating both separately as susceptibility factors. - The authors used the Index of Multiple Deprivation (IMD) as a proxy for family socioeconomic status. Just to clarify, did the study not include more direct assessments of family SES, if not, why not? - Please provide more information about the full available sample from which the 509 toddlers included in the present analyses were abstracted. Based on the language included ("509 toddlers met these inclusion criteria by the date of closure for this analysis", p. 6, line 43), it would seem that more toddlers in this dataset would have met criteria if analyses had waited. - Please provide more information about the sociodemographic information (ranges of observed variables for all continuous variables in Table 1; racial/ethnic breakdown). - Related to the above, I was surprised that the mothers' average age at birth was as high as it was (compared to other birth cohort studies with which I am immediately familiar), and was curious about the range and general distribution. This could be salient if a sizeable proportion of women were of "advanced maternal age"/ "geriatric pregnancies" (pregnancy at age 35 years+, at least in the US), maternal age itself which is a risk for pregnancy outcomes, and warrants discussion. - Equally, if not more importantly, I was surprised that twin and triplet pregnancies were also included, as twin and triplet pregnancies are qualitatively different from singlet pregnancies, and show consistent associations with poorer birth outcomes. These cases should be omitted from analyses, or at minimum,
--	--

	sensitivity analyses should be conducted including and excluding these cases.  - How were infant “clinical characteristics” assessed? - For multiple imputation, the authors indicate that “all variables correlating with the incomplete variables, as well as predictors of the probability of a value being missing” were included in the dataset to aid multiple imputation. Please clarify which variables were considered for inclusion, and which were ultimately included. Please also clarify if this was consistent with the auxiliary variable approach in multiple imputation, and if so, whether a threshold was implemented to determine which variables should be included (i.e., there are diminishing returns when including variables that correlate minimally; see Enders, 2010). Finally, please clarify how it was determined whether predictors were associated with probability of missingness. Were other correlates of probability of missingness also assessed and included, or only putative predictors (i.e., variables collected before missing values were observed)? - Please also include information about reasons for missingness and missing data patterns. I was surprised, for example, that maternal EPDS was the variable with the highest rates of missingness given it was obtained relatively early in the life of the study, and wondered whether there may have been effects related to systematic missingness. - I generally found the results section to be somewhat difficult to follow, and thought it would have been helpful to include subsection headings and/or reminders of which study aims were being tested. It may also be helpful to group preliminary analyses describing preliminary descriptives to explicitly differentiate these from results obtained from primary analyses. - Somewhat related to the above, I also thought it would have been helpful in the discussion to lead with discussions about findings related to primary study aims. I was confused at first to read the discussion opener that mothers of preterm infants did not display more depressive symptoms compared to mothers of term infants, as I did not recall this as a primary study aim (and, on looking back, don’t think it was either). - Given the issues raised with the method as detailed above, I reserve further comment about the discussion at present.
--	---

REVIEWER	Anderson, Cheryl A. The University of Texas at Arlington
REVIEW RETURNED	28-Dec-2021

GENERAL COMMENTS	Well-written and interesting article. I offer a few comments only: 1) sentences should not start with a number (509, 21, 400); 2) reference the classification of scores for the CBCL...your modification or the developer of the tool?; 3) Good discussion and treatment of missing data and data analysis section; 4) sample size overall good but based on smaller samples of mothers with PTB or with depression I am concerned about some of the results without suggesting a limitation to this, especially for statement that subclinical depressive symptoms influence adverse infant affects; 5) I'd like to see the breakout of depressive scores of the EPDS as to subclinical to major depression (>13) and % of very early preterm births, etc..in Table 1; 6) Any thoughts as to why this PPD rate so much lower for this population (>5%)?; 7) Do the cognitive scores from normal to mod-severe reflect other study findings?; 8) Good discussion re link with autism; 9) Good elaboration of limitations and strengths that are mentioned; 10) Timing of PPD
--

	concerning also as 7 weeks can make a big difference with changing of symptoms ----may be overstating some of the findings; 11) references inconsistent in that for some the journal article is in caps and most others not.; 12) very interesting and timely work, thank you for your contribution
--	---

REVIEWER	Berrett, Jenny Cardiff University
REVIEW RETURNED	29-Dec-2021

GENERAL COMMENTS	Thank you to the authors for submitting a well-structured and clear manuscript for review. I found the research to be an interesting read. Below are the points I feel need further explanation: 1. I think it is important throughout your manuscript to keep your language grounded in what you are exploring and have found. Examples of this are found in the abstract: P. 4, line 7 where you state 'mental health' of offspring, however I would argue you consider broader variables than this term. P.4, line 12 – postnatal depressive symptoms were assessed at 'term' – however in your study, they were assessed at different points, and I'm not sure if this is misleading for when EPDS scores were collected. I wonder if this could be reworded as this is an important variable to collect. P. 4, line 36 – “mothers had increased self-reported depressive symptoms...” as you have also stated they exhibit more “maternally-reported behavioural problems”, thus showing this was an important piece of information to include around how the measures were collected. I encourage the authors to check through their manuscript, and consider whether the language such as 'self-reported' or 'maternally reported' is used to ensure findings and conclusions are grounded in their methodology. 2. For your introduction, I wanted to draw your attention, for consideration, of two more recent systematic reviews/analyses: one exploring maternity blues and the other considering the risks of post-natal depression from pre-term delivery, both of which site those papers you have acknowledged in your introduction. Rezaie-Keikhaie, K., Arbabshastan, M. E., Rafiemanesh, H., Amirshahi, M., Ostadkelayeh, S. M., & Arbabisarjou, A. (2020). Systematic Review and Meta-Analysis of the Prevalence of the Maternity Blues in the Postpartum period. Journal of Obstetric, Gynecologic & Neonatal Nursing, 49(2), 127-136. de Paula Eduardo, J. A. F., de Rezende, M. G., Menezes, P. R., & Del-Ben, C. M. (2019). Preterm birth as a risk factor for postpartum depression: A systematic review and meta-analysis. Journal of Affective Disorders, 259, 392-403. 3. I wonder if the methods section could have a little more detail to support replication. For example, what is the justification for inviting toddlers 17-24 months post-expected delivery date, and was the timeframe of the invitation sent e.g. one toddler being invited at 17 months and another at 24 months, based on anything in particular? Was there any specific exclusion criteria apart from not meeting inclusion? a. You clearly outline those variables you included in all analyses, as well as steps taken to account for missing data. It was interesting to see how there was little missing data on some of the variables, with only the EPDS coming close to 15%. I wondered whether this was due to something related to the research
---

	methods/procedures e.g. where parents completed measures? And if there was missing data, were the main reasons for this that are important to mention e.g. why EPDS was higher than other parent-reported measures like CBCL? If considered important, I wondered whether a procedure section outlining how measures were completed or whether there is a need for further explanation such as 'completed in home environment' to support understanding and replication. b. A strength of your study is further trying to differentiate between baby blues and mother's EPDS assessment and EPDS score. It would be helpful in the methods section to clarify what you mean by 'near term-corrected age' on page 9, line 48 – to further help with understanding and future replication. c. Earlier in your transcript, you state a limitation may be "potential shared method variance through parent-completed child behavioural assessments". I wonder if you could add into your variables or analysis section, how you tried to reduce shared method variance e.g. through the measures used, collection of data, analysis etc. 4. A minor revision in the early part of discussion: Principal findings, line 26 – missed 'did not' – "Moreover, gestation age did not influence..." 5. I would like to thank the authors for a clear and transparent limitations section, highlighting key cautions around the research. I hoped to clarify the following points, and wondered whether justifications may be placed in the limitations section: Acknowledging the lack of information regarding e.g. antenatal depression in the study or previous psychiatric history, is important due to the growing literature demonstrating its impact. I hoped to clarify whether this was something that was considered as part of the inclusion and exclusion criteria? And if not, with the growing literature stating that there may be an impact on child development, what the justification may be to not including these outcomes (and adding a sentence around this if possible)? Further, is there justification as to why pre-term EPDS scores were completed later? I wondered if there was an aim to complete an EPDS score within 1 week of birth, as seen in term deliveries. 6. Within future research, I would consider it important to state the need to include more maternal/parental variables. The results of this study need to be interpreted with caution due to variables which have not been included. I agree that the maternal ASD symptomatology should be considered, but also antenatal depression specific to perinatal period, and/or history of mental health difficulties. 7. For table 1, I wonder if it is possible to break down 'preterm', due to research suggesting differences in levels of prematurity e.g. 'very premature' 'late premature' etc. I see from your discussion that you had a small sample for premature, and thus I wonder if your results need to be grounded in 'late prematurity' instead and this acknowledged clearly in your write-up. Conclusion Thank you for submitting for review this piece of research, which I found interesting to read. I felt your limitations section was very thorough and transparent, summarising the cautions I would have
--	--

	when interpreting the results of this study. I do find that some of your statements may need to be further grounded in your methodology, for example, in your conclusion section, line 38, it would be helpful to state “However, we do show that early subclinical maternal postnatal depressive symptoms are associated with behavioural problems in children, on parent-reported measures”. I suggest this, due to the limitation as you pointed out, of how maternal depression may affect reporting observations for mothers. I would also consider re-writing your final sentence of your conclusion, to again ground it in your results, stating that “These findings are of great relevance to child and public health, and further research may strengthen its implications for....”, due to the limitations already mentioned.
--	---

VERSION 1 – AUTHOR RESPONSE

Reviewer: 1

Dr. Betty Lin, University at Albany

Comments to the Author:

The present analyses examined links between maternal postnatal depressive symptoms and toddler adjustment outcomes, and whether preterm birth would heighten susceptibility to maternal postnatal depressive symptoms for better and for worse. There were many study strengths, including the large sample size, longitudinal/ birth cohort design, and use of multiple imputation to address missing data. Additionally, it is readily apparent that the authors were thorough in testing potential alternate explanations (e.g., such as evidenced by testing the effects of time lag of EPDS assessment on EPDS scores). The author also took strides to be transparent about their analytic approach and available data (e.g., such as evidenced by comparisons of analyses using MI and using only complete data). I had some questions and concerns, which I detail below.

We thank Dr Lin for recognising the strengths of the study, as well as our efforts to mitigate limitations inherent to the data. We have numbered comments to allow for cross-references to be cited in our replies. Our page-references are with respect to the marked resubmission.

1. From a conceptual standpoint, my understanding is that the differential susceptibility hypothesis does not regard preterm birth as a susceptibility factor. Instead, there is some evidence that indices of fetal growth (especially low birthweight and intrauterine growth restriction) may represent proxies suggestive of susceptibility (see Pluess & Belsky, 2011, *Development & Psychopathology*; Seckl, 2001, *Molecular & Cellular Endocrinology*). Accordingly, I think it's fine to frame preterm birth as a proxy variable, but would adjust language here for clarity and discuss implications accordingly (i.e., insofar as it is a proxy variable, it is an imperfect capture of putative susceptibility per differential susceptibility).

Thank you for your comments and citations. We understand the differential susceptibility hypothesis to be a theoretical framework, with various clinical characteristics suggested as possible susceptibility factors. As you rightly comment, some researchers have studied low birth weight as such a factor. However, others – such as Hadfield et al (2017)¹ – have studied preterm birth in itself as a susceptibility factor. Indeed, the plasticity of developing neural tissue makes preterm infants a high-risk group for differential susceptibility to external stimuli, as outlined in the review by DeMaster et al (2019).² From a pathophysiological perspective, preterm infants are a distinct clinical group from low-birthweight term infants (which is the group studied by Pluess & Belsky, 2011)³, and there are likely to be different neural/physiological processes underlying their respective hypothesised differential susceptibilities.

Thus, we did not intend to use preterm birth as a proxy variable, but rather as a susceptibility factor in itself. As the majority of babies who are born preterm are also born with a low birthweight,⁴ we used birthweight as a confounder in our models, in order to identify the effect of preterm birth independent of birthweight. The existing literature highlights the different causative factors and long-term outcomes associated with low birthweight secondary to growth restriction in a term baby (i.e. small for gestational age), versus low birthweight secondary to prematurity with lack of time to mature,⁵ with overlap of these two phenomena occurring in certain preterm infants. Indeed, the importance of considering fetal growth restriction in the context of preterm birth has also been highlighted.⁶ Therefore, it was prudent for us to investigate the effect of preterm birth while controlling for birthweight in the same model.

We have expanded our introductory discussion of the differential susceptibility hypothesis and our rationale for studying preterm birth in this context (page 6, lines 177-188), and hope that this removes any confusion over this study's aims.

2. Discussion of background literature in the introduction regarding differential susceptibility in relation to adjustment outcomes felt somewhat overly cursory, particularly regarding potential differential effects of postnatal depressive symptoms on emotional and behavioral adjustment, ASD symptoms, and cognitive development. Related, please note that the relations of postnatal depressive symptoms with cognitive development is not currently included in the stated aims.

Thank you for highlighting the lacking discussion about differential susceptibility and outcomes. We have expanded the introductory paragraph discussing the concept of differential susceptibility, and have included in this the exposures and outcomes studied so far within this theoretical framework (introduction, page 6, lines 177-188). We also highlight that postnatal depressive symptoms have not been studied within a differential susceptibility framework, emphasising the importance and novelty of our own study. Earlier in the introduction (paragraph 2) we explain how child outcomes are associated with maternal depression, and our final introductory paragraph concludes our aim being to conduct a novel study investigating the effect of maternal depressive symptoms on child developmental outcomes, with a focus on a potential differential effect on preterm infants. We hope that the expansion of our discussion of differential susceptibility is acceptable to you.

With regards to cognitive development, this was not one of our primary outcomes and was therefore not included in the stated aims. Rather, child cognition was a covariate that we controlled for in our models of child behavioural outcomes, as our objective was to investigate the association between maternal depressive symptoms and child CBCL and Q-CHAT scores. We repeated the analyses using Bayley-III cognitive composite score as the dependent variable to demonstrate that maternal depressive symptoms were selectively associated with child's indicators of mental health rather than overall cognitive development. We apologise for the misleading description of the cognition analysis in our methods-analysis section and have corrected this (page 11, lines 344-347). We have also specified our key outcomes in the last introductory paragraph, namely behavioural and emotional outcomes, and ASD symptoms.

3. Considering the centrality of preterm birth for the conceptual model, it wasn't clear to me whether gestational age – which was entered in the analyses to reflect preterm birth – was coded as a continuous (gestational age) or dichotomous (preterm or term birth) variable. While I'm generally a proponent of retaining continuous properties of variables for analyses, if the condition of preterm v. term birth is as foundational as the authors suggest given the conceptual framework, then the reliance on the continuous variable v. dichotomizing the variable should be justified.

Thank you for highlighting this confusion. We used gestational age as a continuous variable for our initial models of CBCL and QCHAT. However, in order to investigate the interactive effect of maternal EPDS score and preterm birth, we had to dichotomise the gestational age variable into term/preterm. We have amended our methods analysis section (page 11, lines 337-342) to clarify this.

4. Somewhat related to the above, there were several places throughout the manuscript where the authors were inconsistent with term usage, which made it difficult to distinguish whether terms were intentionally changed (e.g., term birth & gestational age; maternal depression & maternal depressive symptoms) or used interchangeably. For clarity, it would help if the authors used the same terms to describe the same constructs and to additionally consider explicit statements differentiating variations of terms that are salient. Similarly, it would help to clarify if “term corrected” only applied to infants who were born preterm (i.e., before 37 weeks). Finally, I also found it confusing that gestational age was generally viewed as a variable of primary interest but was sometimes described as a “confounder” (e.g., see p. , line 37).

Thank you for outlining the inconsistencies. ‘Gestational age’ is used throughout the methods, as this is the variable we studied. Further, we have defined preterm birth, and have used this throughout the manuscript for consistency. Our introduction discusses postnatal maternal depression as a clinical diagnosis, as per the existing literature. We have added a sentence in our aims (introduction, last paragraph) explaining that we specifically wanted to study maternal depressive symptoms, rather than the dichotomous diagnostic measure of ‘depression’ versus ‘no depression’. We have ensured that any use of ‘maternal depression’ is when describing other studies that investigated this construct, but ‘maternal depressive symptoms’ when in reference to our own study findings. We have explained in the ‘maternal variables’ and ‘analysis’ sections of the methods that maternal EPDS was taken at term-corrected age for preterm infants; we have removed the term from the rest of the manuscript to avoid confusion. We have rephrased the sentence (page 11, line 347) to say ‘variables’ instead of ‘confounders’. We hope that these changes clarify the terms.

5. Also related to the issue of preterm birth as a proxy variable, I don’t understand the rationale for also include birth weight as a covariate in analyses, especially because the latter may represent a better proxy of susceptibility than preterm birth. What were correlations between preterm birth, gestational age, and birth weight? I’m wondering if there may have been issues with multicollinearity. It would help to include correlations among all covariates and key study variables in general, including only one or the other in models, and potentially additionally investigating both separately as susceptibility factors.

Please refer to our previous answer, Point 1. With respect to the comment regarding multicollinearity, we would like to thank the reviewer for querying the collinearity of the variables involved in the analysis models. STATA automatically omits a variable if it identifies a dependency among the independent variables that may hinder the estimation process. No variables were omitted as a result of such dependencies during estimation, indicating that there was no perfect collinearity among our independent variables. As an extra check, we also examined the variance inflation factor (VIF) using the user-written STATA’s command ‘mivif’. The command calculates the VIFs within each imputed dataset and reports the mean VIF for each independent variable over the total number of the imputed datasets. The mean VIFs for the CBCL and QCHAT models were 1.67 and 1.72, respectively, which is not indicative of multicollinearity with our data.

6. The authors used the Index of Multiple Deprivation (IMD) as a proxy for family socioeconomic status. Just to clarify, did the study not include more direct assessments of family SES, if not, why not?

We did not use any other assessments of family SES. The IMD is a widely used and accepted gold-standard epidemiological tool in the UK; example publications using IMD include: Kanel et al, 2021, eNeuro; Little & Nestel, 2017, Lancet.^{7,8} We have expanded our explanation of the IMD in our methods (pages 7, lines 229-233), providing further details on how it encompasses both individual and area-level social risk. Of note, the IMD is also very acceptable to participants, as it relies on postcode rather than the divulging of any personal financial information.

7. Please provide more information about the full available sample from which the 509 toddlers included in the present analyses were abstracted. Based on the language included (“509 toddlers met these inclusion criteria by the date of closure for this analysis”, p. 6, line 43), it would seem that more toddlers in this dataset would have met criteria if analyses had waited.

You are correct in that data collection for the DHCP study is still ongoing. However, the first author had protected academic time and therefore a closure data for this specific analysis had to be drawn at a feasible time-point (26/2/2020).

8. Please provide more information about the sociodemographic information (ranges of observed variables for all continuous variables in Table 1; racial/ethnic breakdown).

We have added ranges for the continuous variables and included data on maternal ethnicity in Table 1.

9. Related to the above, I was surprised that the mothers’ average age at birth was as high as it was (compared to other birth cohort studies with which I am immediately familiar), and was curious about the range and general distribution. This could be salient if a sizeable proportion of women were of “advanced maternal age”/ “geriatric pregnancies” (pregnancy at age 35 years+, at least in the US), maternal age itself which is a risk for pregnancy outcomes, and warrants discussion.

Thank you for raising the factor of advancing maternal age on outcomes. Two of the largest cohort studies investigating maternal mental health and child outcomes, namely the Generation R Study and Avon Longitudinal Study of Parents and Children study, had a mean maternal age of 32 years and 29 years, respectively.⁹ These studies recruited in the 1990s-2000s, and the mean age of childbirth has been on the rise in the UK since then.¹⁰ Thus, maternal age in our sample seems similar to data reported in previous studies and subsequent changing societal trends. Maternal age was normally distributed in our sample, with mean 34.2 years, median 34 years, and mode 33 years; the range has been added to Table 1, showing a maximum maternal age was 52 years. We have amended our discussion to include discussion of these findings (page 18, lines 587-589), especially in light of our sample’s over-representation of preterm births.

10. Equally, if not more importantly, I was surprised that twin and triplet pregnancies were also included, as twin and triplet pregnancies are qualitatively different from singlet pregnancies, and show consistent associations with poorer birth outcomes. These cases should be omitted from analyses, or at minimum, sensitivity analyses should be conducted including and excluding these cases.

The inclusion of twin and triplet pregnancies was carefully considered when planning our data analysis. We did not want to exclude infants of multiple pregnancies because our aim was to investigate a community sample that was as representative of the general population as possible. In addition, twins and triplets are more likely to be born preterm, and removing these from our sample would likely introduce bias into our results. We are aware that having twins or triplets represents a confounder when studying maternal depressive symptoms and child outcomes, and therefore included multiple pregnancy as a variable in our models to account for this. In addition, we are aware of the bias created by including siblings (twins/triplets) in the sample, and accounted for this statistically using clustered robust standard errors. We have expanded our methods section to provide more detail about the number of twin/triplets included in our sample, the number of siblings (page 7, lines 220-221), and the statistical methods used to account for this (page 10, lines 318-320).

11. How were infant “clinical characteristics” assessed?

Infant clinical characteristics were extracted from clinical notes. We have added this detail to the relevant methods section.

12a. For multiple imputation, the authors indicate that “all variables correlating with the incomplete

variables, as well as predictors of the probability of a value being missing” were included in the dataset to aid multiple imputation. Please clarify which variables were considered for inclusion, and which were ultimately included.

Thank you for requesting further clarification. We have now listed the variables that were included in the multiple imputation model in our methods (page 9, lines 295-300).

12b. Please also clarify if this was consistent with the auxiliary variable approach in multiple imputation, and if so, whether a threshold was implemented to determine which variables should be included (i.e., there are diminishing returns when including variables that correlate minimally; see Enders, 2010).

White, Royston and Wood (2009) on page 384-5 argue: *“In practice, one could include in the imputation model all variables that significantly predict the incomplete variable, or whose association with the incomplete variable exceeds some threshold. One might also include any other variables which significantly predict whether the incomplete variable is missing, on the grounds that bias may be avoided if the included variables have a true association with the incomplete variable that fails to reach statistical significance, whereas little loss of precision is likely if the included variables do not predict the incomplete variable.”*¹¹

We acknowledge that a variable should bear an association with the incomplete variable and possibly with the missingness probability. Collins, Schafer and Chi-Ming Kam (2001) state that auxiliary variables are those variables which are included in the analysis solely to improve the performance of the missing data procedure and can be either causes or correlates of the missingness itself, or simply correlated with the incomplete variables, regardless of whether they are related to the factors underlying the missingness.¹² We are aware that, due to the existence of missing data, it is sometimes hard for a correlate to show the association it would have had with the incomplete variable if the sample had been fully observed. For this reason, and coupled with the fact that our analyses estimates can only be conservative, we decided to include maternal age in the imputation model because it was shown to be predictive of the missingness probability.

We have added a sentence in the manuscript stating that we have used a 5% level to define a significant association (page 10, lines 305).

12c. Finally, please clarify how it was determined whether predictors were associated with probability of missingness. Were other correlates of probability of missingness also assessed and included, or only putative predictors (i.e., variables collected before missing values were observed)?

We have added text in the manuscript clarifying how we determined that a variable was associated with the probability of missingness (page 9-10, lines 300-305). This reads: *To assess whether maternal age was predicting the probability of a value being observed, we firstly constructed binary indicators, one for each incomplete variable, that denoted whether the incomplete variable was missing their value (coded 0) or not (coded 1). These indicators then formed the dependent variable in logistic regression models that used maternal age as the independent variable.*

13. Please also include information about reasons for missingness and missing data patterns. I was surprised, for example, that maternal EPDS was the variable with the highest rates of missingness given it was obtained relatively early in the life of the study, and wondered whether there may have been effects related to systematic missingness.

Thank you for highlighting the need for discussion of data missingness. Maternal EPDS in our dataset was the variable with the highest rate of missingness (14.3%). We believe this may be due to the very fact that mothers were asked to complete the EPDS at the time of infant’s MRI scan (as part of the DHCP study) and that some mothers will have had simultaneous childcare duties for siblings, thus limiting their capacity to complete a long questionnaire. We have expanded our discussion of this in the limitations (page 17, lines 569-572).

14. I generally found the results section to be somewhat difficult to follow, and thought it would have been helpful to include subsection headings and/or reminders of which study aims were being tested.

It may also be helpful to group preliminary analyses describing preliminary descriptives to explicitly differentiate these from results obtained from primary analyses.

Thank you for your comments regarding organisation of the results section. We have re-ordered the results section to match the order analyses are presented in the methods, and have used subheadings to improve ease of understanding (pages 11-13).

15. Somewhat related to the above, I also thought it would have been helpful in the discussion to lead with discussions about findings related to primary study aims. I was confused at first to read the discussion opener that mothers of preterm infants did not display more depressive symptoms compared to mothers of term infants, as I did not recall this as a primary study aim (and, on looking back, don't think it was either).

Thank you for highlighting this point. We have re-organised the discussion so that the order of results discussed matches the order that analyses are presented in the methods and results sections (page 14 onwards).

16. Given the issues raised with the method as detailed above, I reserve further comment about the discussion at present.

We look forward to your comments about the discussion in the future.

Reviewer: 2

Dr. Cheryl A. Anderson, The University of Texas at Arlington

Comments to the Author:

Well-written and interesting article.

Thank you for your positive summary.

I offer a few comments only:

1) sentences should not start with a number (509, 21, 400);

Thank you for highlighting this; we have amended the text where applicable.

2) reference the classification of scores for the CBCL...your modification or the developer of the tool?;

Thank you for this question. We used the developer's (ASEBA) references for normal/borderline/clinical ranges. On cross-checking, however, we realise that we had used the given cut-offs for the syndrome scales rather than the less conservative ranges given for the total scale (page 71 in Achenbach & Rescorla, 2000)¹³. We have now corrected this (page 8, lines 260-262), have updated the frequencies in each category accordingly (page 12, lines 374-377), and provided the reference explicitly for the sentence regarding ranges (page 8, lines 262).

3) Good discussion and treatment of missing data and data analysis section;

Thank you for acknowledging the effort made.

4) sample size overall good but based on smaller samples of mothers with PTB or with depression I am concerned about some of the results without suggesting a limitation to this, especially for statement that subclinical depressive symptoms influence adverse infant affects;

Thank you for making this valid point. We have expanded our discussion of these limitations (page 17, lines 564-567), and also included a figure in the supplement showing the distribution of maternal EPDS.

5) I'd like to see the breakout of depressive scores of the EPDS as to subclinical to major depression (>13) and % of very early preterm births, etc..in Table 1;

We have added these sub-groups to Table 1. We have also included a histogram in the Supplement, showing the distribution of maternal EPDS scores.

6) Any thoughts as to why this PPD rate so much lower for this population (>5%)?;

Thank you for this valid question. We have included an additional reference in the discussion, highlighting the previous UK community prevalence of postpartum depression (8.9%), which was assessed using the same EPDS cut-off of 13 at eight weeks postpartum (page 17, line 566). We have also expanded our discussion of the high rate of missingness in the EPDS data (page 17-18,

lines 569-572). One possible explanation for the rate of PPD being relatively low, as discussed in the first paragraph of limitations, is that we did not capture the full extent of postpartum depression in mothers of term infants because the EPDS was completed shortly after birth.

7) Do the cognitive scores from normal to mod-severe reflect other study findings?;

The distribution of the normative sample tested by Bayley shows that 9% had a cognitive composite score <85, and 1% had a score <70.¹⁴ This is not dissimilar to our sample, in which 4.7% had a score between 70-84 (mild impairment), and 1% had a score between 55-69 (moderate impairment). We have summarised this on page 12, lines 377-381.

8) Good discussion re link with autism;

Thank you.

9) Good elaboration of limitations and strengths that are mentioned;

Thank you.

10) Timing of PPD concerning also as 7 weeks can make a big difference with changing of symptoms ----may be overstating some of the findings;

Thank you for recognising this, and we absolutely agree with you. We tried to investigate the impact of this time-lag on maternal EPDS scores, as shown in Supplement Table 4 and 5; these post-hoc analyses provided reassurance that our assessments of mothers of term-born infants were not inflated by the common, temporary symptoms of baby blues. However, it is possible that we are not observing the full severity of depressive symptoms in mothers of terms infants, as they were assessed before postpartum depression typically presents. We have discussed this as our first limitation in the discussion page 17, line 552-567).

11) references inconsistent in that for some the journal article is in caps and most others not.; Apologies for the formatting errors; we have corrected these.

12) very interesting and timely work, thank you for your contribution
Thank you for your favourable feedback.

Reviewer: 3

Dr. Jenny Berrett, Cardiff University, Swansea Bay University Health Board

Comments to the Author:

Thank you to the authors for submitting a well-structured and clear manuscript for review. I found the research to be an interesting read.

Thank you for this positive feedback.

Below are the points I feel need further explanation:

1. I think it is important throughout your manuscript to keep your language grounded in what you are exploring and have found. Examples of this are found in the abstract:

P. 4, line 7 where you state 'mental health' of offspring, however I would argue you consider broader variables than this term.

We have changed the term to 'behavioural outcomes' throughout the manuscript, to more accurately reflect what was tested in our study.

P.4, line 12 – postnatal depressive symptoms were assessed at 'term' – however in your study, they were assessed at different points, and I'm not sure if this is misleading for when EPDS scores were collected. I wonder if this could be reworded as this is an important variable to collect.

We have amended the terminology and now use 'term-equivalent' to describe the timing of EPDS assessment. We have included an explanatory statement in the methods to explain the meaning of 'term-equivalent' i.e. that mothers of term infants were assessed shortly after birth, whereas mothers of preterm infants were assessed at term-corrected age (page 8, line 236-238).

P. 4, line 36 – "mothers had increased self-reported depressive symptoms..." as you have also stated they exhibit more "maternally-reported behavioural problems", thus showing this was an important piece of information to include around how the measures were collected.

We specify in the methods (page 8, line 238-242) that the EPDS "is a 10-item screening questionnaire completed by mothers"; hence, mothers self-reported their depressive symptoms. Under the 'child

variables' section of the methods (page 8, line 254), we also specify that the CBCL is "a parent-completed 100-item questionnaire"; hence, children's behavioural problems are parentally-reported.

I encourage the authors to check through their manuscript, and consider whether the language such as 'self-reported' or 'maternally reported' is used to ensure findings and conclusions are grounded in their methodology.

Thank you for highlighting this point. We have re-framed our conclusions, both in the abstract and discussion, with this in mind, and have been more consistent with our description of depressive symptoms and behavioural outcomes as maternally or parentally-reported, respectively. We discuss the limitations of potential common method variance in our discussion (page 19, lines 604-609).

2. For your introduction, I wanted to draw your attention, for consideration, of two more recent systematic reviews/analyses: one exploring maternity blues and the other considering the risks of post-natal depression from pre-term delivery, both of which cite those papers you have acknowledged in your introduction.

Rezaie-Keikhaie, K., Arbabshastan, M. E., Rafiemanesh, H., Amirshahi, M., Ostadkelayeh, S. M., & Arbabisarjou, A. (2020). Systematic Review and Meta-Analysis of the Prevalence of the Maternity Blues in the Postpartum period. *Journal of Obstetric, Gynecologic & Neonatal Nursing*, 49(2), 127-136.

de Paula Eduardo, J. A. F., de Rezende, M. G., Menezes, P. R., & Del-Ben, C. M. (2019). Preterm birth as a risk factor for postpartum depression: A systematic review and meta-analysis. *Journal of Affective Disorders*, 259, 392-403.

Thank you for including the references for these two interesting papers. We have included reference to them in our introduction.

3. I wonder if the methods section could have a little more detail to support replication. For example, what is the justification for inviting toddlers 17-24 months post-expected delivery date, and was the timeframe of the invitation sent e.g. one toddler being invited at 17 months and another at 24 months, based on anything in particular? Was there any specific exclusion criteria apart from not meeting inclusion?

Thank you for seeking clarification on this. We have amended the methods to clarify that children were invited to attend follow-up at 18 months post-expected delivery date, but that the exact appointment time depended on family availability (page 7, lines 213-216). Our results section specifies the age-range at which children were then seen. We have also included further details about the DHCP exclusion criteria (page 7, lines 211-213). There were no additional exclusion criteria, apart from not meeting inclusion, for our follow-up study.

a. You clearly outline those variables you included in all analyses, as well as steps taken to account for missing data. It was interesting to see how there was little missing data on some of the variables, with only the EPDS coming close to 15%. I wondered whether this was due to something related to the research methods/procedures e.g. where parents completed measures? And if there was missing data, were the main reasons for this that are important to mention e.g. why EPDS was higher than other parent-reported measures like CBCL? If considered important, I wondered whether a procedure section outlining how measures were completed or whether there is a need for further explanation such as 'completed in home environment' to support understanding and replication. Thank you for raising this point, which was also shared by your fellow reviewer colleague. We have included further discussion about the high rate of missingness of maternal EPDS in our discussion (page 17, lines 569-572). We have also expanded our methods section to provide more information about the testing environment (page 8, lines 242-246).

b. A strength of your study is further trying to differentiate between baby blues and mother's EPDS

assessment and EPDS score. It would be helpful in the methods section to clarify what you mean by 'near term-corrected age' on page 9, line 48 – to further help with understanding and future replication.

Thank you for highlighting the confusion around the terminology and timing. We have removed 'near term-corrected age' from the manuscript and been more consistent with the use of 'term-equivalent'. We have explained the meaning of this, describing the timing of testing for mothers of preterm and term infants, in our methods 'maternal variables' section (page 8, lines 236-238).

c. Earlier in your transcript, you state a limitation may be "potential shared method variance through parent-completed child behavioural assessments". I wonder if you could add into your variables or analysis section, how you tried to reduce shared method variance e.g. through the measures used, collection of data, analysis etc.

Thank you for your comment. In order to decrease participants' burden we designed a relatively short assessment protocol which included behavioural assessments of toddlers based on parent-completed measures, of which the majority were maternally-completed. We tried to minimise common method bias by selecting assessment tools with good cross-informant agreement, such as the CBCL;¹³ we have amended our methods to explain this advantage of the CBCL. However, we are aware of and discuss the possible common method variance as an unavoidable limitation (page 18-19, lines 604-609), and call for further studies to use independent, objective measures (page 19, lines 626-628).

4. A minor revision in the early part of discussion: Principal findings, line 26 – missed 'did not' – "Moreover, gestation age did not influence..."

Thank you for spotting this typing error. We have re-written this section of the manuscript in response to one of your reviewer colleagues.

5. I would like to thank the authors for a clear and transparent limitations section, highlighting key cautions around the research. I hoped to clarify the following points, and wondered whether justifications may be placed in the limitations section:

Acknowledging the lack of information regarding e.g. antenatal depression in the study or previous psychiatric history, is important due to the growing literature demonstrating its impact. I hoped to clarify whether this was something that was considered as part of the inclusion and exclusion criteria? And if not, with the growing literature stating that there may be an impact on child development, what the justification may be to not including these outcomes (and adding a sentence around this if possible)?

We thank the reviewer for this comment and we acknowledge that the lack of comprehensive information on parental perinatal mental health and psychiatric history is a limitation of this study. The data used for this analysis are derived from the Developing Human Connectome Project (DHCP), which is a neuroimaging-focused project aimed at creating a dynamic map of human brain connectivity during the fetal and neonatal period; it was not designed to answer in-depth questions about maternal mental health. The DHCP questionnaires completed by mothers at infants' term-equivalent age included questions regarding history of mental health conditions, but data collected on this topic were very limited and not amenable to further analysis. We agree with the reviewer that, in these circumstances, it is important to state that the observed results need to be interpreted with caution, as they may be driven by variables that were not measured (e.g., maternal anxiety); we emphasise this caution on page 18, lines 578-581. We have also included a suggestion for future research to consider mental health more in-depth, investigating the overlapping and potentially additive effects of maternal and paternal mental health, as well as socioeconomic and environmental factors on child outcomes (page 19, lines 624-626).

Further, is there justification as to why pre-term EPDS scores were completed later? I wondered if there was an aim to complete an EPDS score within 1 week of birth, as seen in term deliveries.

Thank you for querying this point. Our justification for using EPDS scores at term-equivalent age for all participants was because all infants and mothers attended for the child's MRI at term-equivalent age and thus all were requested to complete the EPDS at that time-point. A small proportion of preterm infants did have MRI scans and maternal EPDS performed earlier (so twice: soon after preterm birth and at term-equivalent age), but this was not consistent. We have analysed the data available for mothers of preterm infants who completed multiple EPDS assessments, which shows no change in EPDS score between time-points ($t(59)=-1.08$, $p=0.29$). This provides us with additional confidence that EPDS responses were stable in the first few months postnatally, although we acknowledge the time-lag between birth and EPDS assessment for preterm participants to be a significant limitation and elaborate on this in our discussion (page 17, lines 552-567).

6. Within future research, I would consider it important to state the need to include more maternal/parental variables. The results of this study need to be interpreted with caution due to variables which have not been included. I agree that the maternal ASD symptomatology should be considered, but also antenatal depression specific to perinatal period, and/or history of mental health difficulties.

Thank you, we completely agree and have included this in the future research section (page 19, lines 624-626).

7. For table 1, I wonder if it is possible to break down 'preterm', due to research suggesting differences in levels of prematurity e.g. 'very premature' 'late premature' etc.

We have now included this information in Table 1.

I see from your discussion that you had a small sample for premature, and thus I wonder if your results need to be grounded in 'late prematurity' instead and this acknowledged clearly in your write-up.

Thank you raising this discussion point. The distribution of preterm groups, which is now published in Table 1, shows that about half of preterm infants were late preterm, and the other half split between very and extremely preterm. Therefore, we would argue that our findings with respect to prematurity are not grounded solely in 'late prematurity'. However, it is true that the number of preterm participants is perhaps insufficient to identify any differential effect of prematurity overall, and we have now included this in our discussion (page 18, lines 587-592).

Conclusion

Thank you for submitting for review this piece of research, which I found interesting to read. I felt your limitations section was very thorough and transparent, summarising the cautions I would have when interpreting the results of this study. I do find that some of your statements may need to be further grounded in your methodology, for example, in your conclusion section, line 38, it would be helpful to state "However, we do show that early subclinical maternal postnatal depressive symptoms are associated with behavioural problems in children, on parent-reported measures". I suggest this, due to the limitation as you pointed out, of how maternal depression may affect reporting observations for mothers. I would also consider re-writing your final sentence of your conclusion, to again ground it in your results, stating that "These findings are of great relevance to child and public health, and further research may strengthen its implications for...", due to the limitations already mentioned.

Thank you for highlighting the need to re-frame the conclusion to better reflect our data, and thank you for the excellent suggestions. They have been incorporated.

References

1. Hadfield K, O'Brien F, Gerow A. Is level of prematurity a risk/plasticity factor at three years of age? *Infant Behav Dev.* 2017;47:27-39. doi:10.1016/j.infbeh.2017.03.003

2. DeMaster D, Bick J, Johnson U, Montroy JJ, Landry S, Duncan AF. Nurturing the preterm infant brain: leveraging neuroplasticity to improve neurobehavioral outcomes. *Pediatr Res.* 2019;85(2):166-175. doi:10.1038/s41390-018-0203-9
3. Pluess M, Belsky J. Prenatal programming of postnatal plasticity? *Dev Psychopathol.* 2011;23(1):29-38. doi:10.1017/S0954579410000623
4. Blencowe H, Krasevec J, de Onis M, et al. National, regional, and worldwide estimates of low birthweight in 2015, with trends from 2000: a systematic analysis. *Lancet Glob Health.* 2019;7(7):e849-e860. doi:10.1016/S2214-109X(18)30565-5
5. World Health Organization (WHO). Global nutrition targets 2025: low birth weight policy brief. Published online 2014.
6. Zeitlin JA, Ancel PY, Saurel-Cubizolles MJ, Papiernik E. Are risk factors the same for small for gestational age versus other preterm births? *Am J Obstet Gynecol.* 2001;185(1):208-215. doi:10.1067/mob.2001.114869
7. Kanel D, Vanes LD, Pecheva D, et al. Neonatal White Matter Microstructure and Emotional Development during the Preschool Years in Children Who Were Born Very Preterm. *eNeuro.* 2021;8(5). doi:10.1523/ENEURO.0546-20.2021
8. Little E, Nestel P. Association of deprivation with overweight and obesity among primary school children in England: an ecological cross-sectional study. *The Lancet.* 2017;390:S59. doi:10.1016/S0140-6736(17)32994-X
9. Van Batenburg-Eddes T, Brion MJ, Henrichs J, et al. Parental depressive and anxiety symptoms during pregnancy and attention problems in children: a cross-cohort consistency study. *J Child Psychol Psychiatry.* 2013;54(5):591-600. doi:10.1111/jcpp.12023
10. Office for National Statistics. Birth characteristics in England and Wales: 2020. Published January 13, 2022. Accessed February 5, 2022. <https://www.ons.gov.uk/peoplepopulationandcommunity/birthsdeathsandmarriages/livebirths/bulletins/birthcharacteristicsinenglandandwales/2020#birth-characteristics>
11. White IR, Royston P, Wood AM. Multiple imputation using chained equations: Issues and guidance for practice. *Stat Med.* 2011;30(4):377-399. doi:https://doi.org/10.1002/sim.4067
12. Collins LM, Schafer JL, Kam CM. A comparison of inclusive and restrictive strategies in modern missing data procedures. *Psychol Methods.* 2001;6(4):330-351.
13. Achenbach TM, Rescorla LA. *Manual for the ASEBA Preschool Forms & Profiles.*; 2000.
14. Bayley N. *Bayley Scales of Infant and Toddler Development - Third Edition.* 3rd ed. Harcourt Assessment; 2006.

VERSION 2 – REVIEW

REVIEWER	Lin, Betty University at Albany
REVIEW RETURNED	28-Mar-2022

GENERAL COMMENTS	I appreciate the authors clarification and elaboration in response to my earlier comments. I especially appreciate the modifications the authors made to the discussion, and think they do a particularly nice job discussing limitations within which to contextualize study results. I have a several remaining questions/comments – many of which represent elaborations or additional questions based the authors responses to my original comments. Please elaborate further in the introduction about why preterm birth and low birthweight should be considered as independent susceptibility factors. Apologies if my original comment about the need for elaboration about differential susceptibility with respect to the outcomes of
--

	interest was unclear. Please elaborate about why we would expect preterm birth to heighten susceptibility to the effects of maternal depressive symptoms on infant emotional and behavioral problems. I'm particularly concerned about the focus on ASD symptoms as an outcome associated with postnatal caregiving moods or behaviors, which is in contrast to current understanding that it is biologically (and not environmentally) based, and could be problematic (i.e., I'm thinking about historical misattributions of ASD to a lack of parent warmth and sensitivity/ "refrigerator mothers/parents"). The authors likewise seem to acknowledge in the discussion that there is no current clear etiological role of maternal depression in the development of ASD, which would seem to suggest that it was contraindicated to test to begin with. The empirical basis for testing this question at all should be explicitly stated in the introduction or the QCHAT should be dropped from analyses altogether. Thanks for clarifying when preterm birth was treated as a continuous v. dichotomous variable in analyses. However, I'm still not sure I understand the conceptual rationale for why the authors chose to treat them as such. Since EPDS was only administered at term-corrected age for preterm infants, I assume this means the authors think there is something qualitatively distinct about infants born < 37 weeks compared to those born > 37 weeks versus that differences between preterm and term infants were incremental and continuous. If this is the case, the justification for this hard cut-off should be explicitly stated (e.g., clinical relevance?), and all analyses should treat the preterm variable as dichotomous. Thanks for adding ranges. Some ranges are quite large (e.g., birthweight 450-4750g, age range 17-52 years). Were there many extremely/very low birthweight infants, or is the lower range somewhat anomalous? Were variables generally inspected for normality and extreme values? If so, how were these handled? Similar to above, I'm surprised at how broad the age range is of women who participated. Were there many women who gave birth in the 40s and 50s? I wonder if there may be qualitative developmental differences as a function of age. Assuming there aren't any age outliers (i.e., that the participant(s) aged 52 years weren't overly anomalous, I appreciate the authors' point about how the broadly sampled age range could be more representative of today's society, and think additional (brief) discussion about limitations associated with the focus on a broad age range/ need for future research evaluating possible age moderation would help to round out this point. Are sample demographics representative of the geographic area from which data were collected? I appreciate the consideration of twin/triplet and siblings in statistical analyses. I defer to the editor about importance, but I personally would feel more confident if sensitivity analyses (i.e., analyses with and without the twin/triplet and siblings) were presented. RE: missing data screening, it sounds like besides key study variables, maternal age was the only variable considered for inclusion as an auxiliary variable during multiple imputation. Is this correct? If so, why weren't any other sociodemographic or conceptually related variables considered for inclusion? Please also note that, per Enders (2010), the recommended threshold is a correlation of > .30, and is not based on statistical significance threshold. This should be corrected or an alternate citation supporting a $p = .05$ threshold should be provided.
--	--

	Thanks for clarifying about possible reasons why maternal EPDS may have had the highest rates of missing data. Please comment on patterns of missingness on key study variables in general, and whether any sociodemographic variables were correlated with missingness. I found the results of the interaction effect to be somewhat confusing and non-intuitive to follow as presented. I would suggest removing the subheading and combining this section with the section describing the results of the full regression term, and explicitly stating that the interaction term was not significant, including in the abstract. There doesn't seem to be a need to present results of t-tests exploring the non-significant interaction terms; this is true for the abstract, as well. It should also be noted in the analytic plan that the regressions were run twice each – once with and without the interaction terms. Please remove the section “Association between maternal EPDS score and toddler cognitive outcomes” – it seems to imply that these associations are of central interest, though the authors note that cognitive outcomes were only included as a covariate.
--	--

VERSION 2 – AUTHOR RESPONSE

Reviewer 1

I appreciate the authors clarification and elaboration in response to my earlier comments. I especially appreciate the modifications the authors made to the discussion, and think they do a particularly nice job discussing limitations within which to contextualize study results. I have a several remaining questions/comments – many of which represent elaborations or additional questions based the authors responses to my original comments.

Please elaborate further in the introduction about why preterm birth and low birthweight should be considered as independent susceptibility factors.

In the introduction we specify that low birthweight in term infants (i.e. small for gestational age, SGA) is pathophysiologically distinct from preterm birth as a cause of low birthweight. We have expanded this section of the introduction to provide examples, with additional citations that the reviewer is welcome to explore, of how the pathophysiology of preterm birth can differ from that of SGA (page 6, line 183-187). We feel it is not within the scope of our manuscript's introduction to go into even further detail of this complex topic, but hope that the signposting to relevant further reading for our readers will be viewed favourably.

Of course, as mentioned in the introduction, there can be overlap between SGA and PTB. However, it was not our aim in this paper to dissect the different effects of preterm birth versus SGA on outcomes – although this would be an interesting avenue of further work; instead, we investigated the outcomes of a cohort of preterm infants compared to term infants. We hope that the additional information adequately explains our rationale for distinguishing preterm birth from low birthweight.

Apologies if my original comment about the need for elaboration about differential susceptibility with respect to the outcomes of interest was unclear. Please elaborate about why we would expect

preterm birth to heighten susceptibility to the effects of maternal depressive symptoms on infant emotional and behavioral problems.

We have re-written the final paragraph of our introduction, which states our aims, in order to draw more reference to the science discussed earlier in the introduction. We have framed our hypothesis in the context of the prior research around the differential susceptibility model of prematurity, which is discussed in detail in paragraph four of the introduction, and explained that we wished to explore the effect of maternal depressive symptoms precisely because this had not yet been studied as a stimulus to which preterm infants may be differentially susceptible to. We believe we have satisfactorily justified the scientific rationale of our novel hypothesis in the preceding introductory paragraphs.

I'm particularly concerned about the focus on ASD symptoms as an outcome associated with postnatal caregiving moods or behaviors, which is in contrast to current understanding that it is biologically (and not environmentally) based, and could be problematic (i.e., I'm thinking about historical misattributions of ASD to a lack of parent warmth and sensitivity/ "refrigerator mothers/parents"). The authors likewise seem to acknowledge in the discussion that there is no current clear etiological role of maternal depression in the development of ASD, which would seem to suggest that it was contraindicated to test to begin with. The empirical basis for testing this question at all should be explicitly stated in the introduction or the QCHAT should be dropped from analyses altogether.

Thank you for reiterating your concern. We used the Quantitative Checklist for Autism in Toddlers (Q-CHAT) as an additional behavioural screening tool to broaden our exploration of mental health outcomes in toddlers. As our aim was to investigate behavioural problems in childhood, we felt it was important to include multiple outcome measures. We acknowledge that the Q-CHAT has a low positive predictive value for autism and therefore is better viewed as a measure of toddler behaviour, not specific to ASD. We have now further explained this at multiple points in the manuscript. We have also removed all reference of 'ASD traits' and instead present our Q-CHAT results as an indicator of childhood behavioural problems.

Thanks for clarifying when preterm birth was treated as a continuous v. dichotomous variable in analyses. However, I'm still not sure I understand the conceptual rationale for why the authors chose to treat them as such. Since EPDS was only administered at term-corrected age for preterm infants, I assume this means the authors think there is something qualitatively distinct about infants born < 37 weeks compared to those born > 37 weeks versus that differences between preterm and term infants were incremental and continuous. If this is the case, the justification for this hard cut-off should be explicitly stated (e.g., clinical relevance?), and all analyses should treat the preterm variable as dichotomous.

We used a continuous measure of gestational age as a covariate in our main analyses because we acknowledge that development occurs on a continuum. Thus, to answer your question directly, we do not believe there is something qualitatively distinct about infants born immediately before versus immediately after 37 weeks gestation, and thus find our main results using a continuous measure of gestational age more insightful. Our interaction model, however, required us to dichotomise the gestational age variable in order to statistically test the interactive effect; thus, in light of the WHO definition of prematurity being <37 weeks, we used this cut-off to allow statistical testing of our hypothesis. This decision to dichotomise the gestational age variable for that specific analysis was made out of statistical methodological necessity. We prefer to present our main/remaining analyses with a continuous gestational variable, for reasons explained above.

Thanks for adding ranges. Some ranges are quite large (e.g., birthweight 450-4750g, age range 17-52 years). Were there many extremely/very low birthweight infants, or is the lower range somewhat anomalous? Were variables generally inspected for normality and extreme values? If so, how were these handled?

Conditional normality was inspected in the complete-case analyses using QQ plots of the residuals of the models. Initially we fit the model with all the data in, constructed the residuals and examined the QQ plot. "Extreme" values were then removed, the models were re-fitted without these values, and new QQ plots of residuals were constructed again to check if there were any new "extreme" values. This process was repeated as many times as needed to remove "extreme" values. During this process, the resulting estimates from the models were being examined as to whether they had substantially changed. We found that the removal of "extreme" values did not make any difference to the estimated parameters and hence decided to keep all the data in.

Similar to above, I'm surprised at how broad the age range is of women who participated. Were there many women who gave birth in the 40s and 50s? I wonder if there may be qualitative developmental differences as a function of age. Assuming there aren't any age outliers (i.e., that the participant(s) aged 52 years weren't overly anomalous, I appreciate the authors' point about how the broadly sampled age range could be more representative of today's society, and think additional (brief) discussion about limitations associated with the focus on a broad age range/ need for future research evaluating possible age moderation would help to round out this point.

Thank you, we have included further brief discussion of the limitation of maternal age.

Are sample demographics representative of the geographic area from which data were collected? I appreciate the consideration of twin/triplet and siblings in statistical analyses. I defer to the editor about importance, but I personally would feel more confident if sensitivity analyses (i.e., analyses with and without the twin/triplet and siblings) were presented.

Our sample was ethnically representative of the surrounding geographical area. Analysis of our sample's IMD scores (as a measure of deprivation) showed that our sample was generally less deprived than the surrounding areas, as well as the UK as a whole. This recruitment bias reflects trends observed in other UK longitudinal studies (Boyd et al., 2013). We have added this information to the methods section (page 7, lines 232 and 234).

RE: missing data screening, it sounds like besides key study variables, maternal age was the only variable considered for inclusion as an auxiliary variable during multiple imputation. Is this correct?

Yes, maternal age was the only additional variable that was included in the imputation model on top of all other variables that were included in the substantive model of interest.

If so, why weren't any other sociodemographic or conceptually related variables considered for inclusion?

Maternal age and IMD were the only sociodemographic variables available for analysis, and were both used in our imputation models.

Please also note that, per Enders (2010), the recommended threshold is a correlation of $> .30$, and is not based on statistical significance threshold. This should be corrected or an alternate citation supporting a $p = .05$ threshold should be provided.

We would like to thank the reviewer for this insightful comment. We agree that the choice of auxiliary variables should not be based on a strict statistical significance threshold. For this reason, we removed from the text the sentence “We used a 5% level to define a significant association”, in order to avoid a misleading use of p-values.

To explore the role of a variable as an auxiliary variable in an imputation model, we start by exploring the strength of the association of clinically meaningful variables with the incomplete variables of interest, either in correlation tests, or in simple regression models using the incomplete variable as the dependent variable and the variable in question as the independent variable. The variables we choose as auxiliary are associated with the incomplete variable and may or may not predict missingness in the incomplete variable of interest. The latter is explored in logistic models after constructing binary indicators of whether a variable is missing or not. Then, depending on model stability, we may add, if available, a few more variables that predict whether a value in the incomplete variable is missing or not, based on the strength of the association seen in the odds ratios and the confidence intervals within the logistic models. We aim for the imputation model to be at least as elaborate as the substantive analysis model of interest. For this study, we had maternal age as the only available variable to examine, and it was strongly associated with both CBCL and Q-CHAT.

Thanks for clarifying about possible reasons why maternal EPDS may have had the highest rates of missing data. Please comment on patterns of missingness on key study variables in general, and whether any sociodemographic variables were correlated with missingness.

IMD was correlated with EPDS missingness, and was included in both the imputation and analysis models.

Please see the table below showing the patterns of missingness in the incomplete variables.

Pattern	IMD rank	Q-CHAT	CBCL	Maternal BMI	Maternal EPDS	Number	Percentage of total
1	✓	✓	✓	✓	✓	400	79
2	✓	✓	✓	✓	·	68	13
3	✓	✓	✓	·	✓	25	5
4	✓	·	·	✓	✓	7	1
5	·	✓	✓	✓	✓	3	<1
6	✓	·	·	✓	·	2	<1
7	✓	✓	✓	·	·	2	<1
8	✓	✓	·	✓	·	1	<1
9	✓	✓	·	✓	✓	1	<1
Total						509	100%

Pattern of missing values in the dataset: ✓=observed, · = missing

I found the results of the interaction effect to be somewhat confusing and non-intuitive to follow as presented. I would suggest removing the subheading and combining this section with the section describing the results of the full regression term, and explicitly stating that the interaction term was not significant, including in the abstract. There doesn't seem to be a need to present results of t-tests exploring the non-significant interaction terms; this is true for the abstract, as well. It should also be noted in the analytic plan that the regressions were run twice each – once with and without the interaction terms.

Thank you for these comments, we have incorporated them into the manuscript.

Please remove the section “Association between maternal EPDS score and toddler cognitive outcomes” – it seems to imply that these associations are of central interest, though the authors note that cognitive outcomes were only included as a covariate.

We have removed this section from the results as requested, as well as the associated text in the methods and Table 4 of the results.

References

Boyd, A., Golding, J., Macleod, J., Lawlor, D. A., Fraser, A., Henderson, J., Molloy, L., Ness, A., Ring, S., & Davey Smith, G. (2013). Cohort Profile: The ‘Children of the 90s’—the index offspring of the Avon Longitudinal Study of Parents and Children. *International Journal of Epidemiology*, *42*(1), 111–127. <https://doi.org/10.1093/ije/dys064>

VERSION 3 – REVIEW

REVIEWER	Lin, Betty University at Albany
REVIEW RETURNED	11-Jul-2022

GENERAL COMMENTS	I appreciate the authors thorough and thoughtful responses to my concerns, and feel they have satisfactorily addressed most concerns quite well. I have two remaining comments: It sounds like the authors engaged a thorough approach to screening for extreme values; this is a study strength, and also helps to address concerns that results may be driven by extreme values. I'd suggest the authors include this information in the method and note that they ran sensitivity analyses with and without extreme values, but because results did not differ, they present the results from the full sample here. RE: criteria for auxiliary variable inclusion, it's a little hard to discern what thresholds the authors used for auxiliary variable screening; if they relied on a $p=.05$ threshold, this is what they should report. However, it seems concerning that the approach taken is in contrast to current recommendations for missing data screening. I defer to the editor about whether it would be more
--

	appropriate for the authors to detail their approach including thresholds used in the manuscript or to re-run analyses using the recommended thresholds.
--	--

VERSION 3 – AUTHOR RESPONSE

Reviewer 1

I appreciate the authors thorough and thoughtful responses to my concerns, and feel they have satisfactorily addressed most concerns quite well. I have two remaining comments:

It sounds like the authors engaged a thorough approach to screening for extreme values; this is a study strength, and also helps to address concerns that results may be driven by extreme values. I'd suggest the authors include this information in the method and note that they ran sensitivity analyses with and without extreme values, but because results did not differ, they present the results from the full sample here.

Thank you for this valuable suggestion. We have now included details of our sensitivity analysis with and without extreme values in the methods (page 10).

RE: criteria for auxiliary variable inclusion, it's a little hard to discern what thresholds the authors used for auxiliary variable screening; if they relied on a $p=.05$ threshold, this is what they should report. However, it seems concerning that the approach taken is in contrast to current recommendations for missing data screening. I defer to the editor about whether it would be more appropriate for the authors to detail their approach including thresholds used in the manuscript or to re-run analyses using the recommended thresholds.

We'd like to thank the reviewer for this comment. We have added some text in the main manuscript explaining that we have used a threshold level of 20% when choosing the auxiliary variables for our imputation models. This can be found on page 9 and reads: "*In the imputation models we also included variables that were associated with the incomplete variables at the 20% level. As such, maternal age was included in the imputation model because it was found to be a significant predictor of the total CBCL raw score ($p=0.001$), the Q-CHAT score ($p=0.021$) and EPDS score ($p=0.122$) when it was included as an independent variable in regression models.*"

VERSION 4 – REVIEW

REVIEWER	Lin, Betty University at Albany
REVIEW RETURNED	08-Aug-2022
GENERAL COMMENTS	I have no remaining comments beyond those provided previously. Thanks for the opportunity to participate in this review!